# Learning Transferable Reward for Query Object Localization with Policy Adaptation

**Tingfeng Li**[†][§][*], **Shaobo Han**[†], **Martin Renqiang Min**[†], **Dimitris N. Metaxas**[§]
[†]NEC Labs America, [§]Department of Computer Science, Rutgers University
{tl601,dnm}@cs.rutgers.edu, {shaobo,renqiang}@nec-labs.com

## Abstract

We propose a reinforcement learning based approach to *query object localization*, for which an agent is trained to localize objects of interest specified by a small exemplary set. We learn a transferable reward signal formulated using the exemplary set by ordinal metric learning. Our proposed method enables test-time policy adaptation to new environments where the reward signals are not readily available, and outperforms fine-tuning approaches that are limited to annotated images. In addition, the transferable reward allows repurposing the trained agent from one specific class to another class. Experiments on corrupted MNIST, CU-Birds, and COCO datasets demonstrate the effectiveness of our approach [1].

## 1 Introduction

There is increasing interest in designing machine learning models that can be adapted to unseen environments or repurposed for new tasks, with only minimal human guidance (Vinyals et al., 2016; Snell et al., 2017; Finn et al., 2017; Sun et al., 2020; Hansen et al., 2021). To this end, a small set of examples can not only serve the purpose of implicitly generalizing a trained model to a new environment during training time, but also enable the learner to update its learning objectives during test time.

In this paper, we focus on a reinforcement learning (RL) formulation to the problem of *query object localization*, in which an agent is trained to localize the target object specified by a small set of exemplary images. Instead of using fixed bounding box proposals, our vision-based agent can be viewed as a proactive information gatherer (Guo, 2003), which actively interacts with an image environment; Furthermore, it follows a class-specific localization policy, and thus is more suitable for aerial imagery (Xia et al., 2018), robotic manipulation (Kalashnikov et al., 2018), or embodied AI (Savva et al., 2019) tasks.

During the test time of query object localization, the queried object class to localize may be novel, or the background environment may undergo substantial changes, hindering the applicability of class-agnostic agents with a fixed policy. In addition, in standard RL settings for object localization, the reward signal is often available, so fine-tuning methods (Julian et al., 2020) can effectively adapt agents to new environments and yield improved performance; On the contrary, in query object localization, the reward signal is not available at test time, because the bounding box annotations are to be found by the localization agent on test images.

To address these problems, we propose an ordinal metric learning based framework for learning an implicitly transferable reward signal defined with a small exemplary set. An ordinal embedding network is pre-trained with data augmentation under a loss function designed to be relevant to the RL task. The reward signal allows explicit updates of the controller in the policy network with continual training during test time. Compared to fine-tuning approaches, the agent can get exposed to new environments more extensively with unlimited usage of test images. Informed by the exemplary set precisely, the agent is adapted to the changes of the localization target.

Our contributions in this paper are summarized as follows: (1) We propose a novel ordinal metric learning based RL framework to learn a transferable reward signal defined with a small exemplary

---

[*]Work done as an intern at NEC Labs America
[1]Code available at https://github.com/litingfeng/Localization-by-OrdEmbed

set for query object localization; (2) Our RL framework enables test-time policy adaptation to new environments where there is no readily available reward signal; (3) Our learned transferable reward allows repurposing a trained agent from localizing one specific class to another; (4) Extensive experiments on several datasets demonstrate the effectiveness of our proposed approach.

## 2 RELATED WORK

### 2.1 OBJECT LOCALIZATION

Compared to bounding-box regression approaches (Redmon et al., 2016; Ren et al., 2016), deep RL based object localization approaches (Caicedo & Lazebnik, 2015; Jie et al., 2016) have the advantage of being region-proposal free, with customized search paths for each image environment. The specificity of an agent purely depends on the classes of bounding-boxes used in the reward. They can be made class-specific (Caicedo & Lazebnik, 2015), but the agent for each class would need to be trained separately.

Despite the rise of crowdsourcing platforms, obtaining ample amount of bounding-box annotations remains costly and error-prone. Furthermore, the quality of annotations often varies, and precise annotations for certain object class may require special expertise from annotators. The emergence of weakly supervised object localization (WSOL) (Song et al., 2014; Oquab et al., 2015; Zhou et al., 2016) methods alleviates the situation, in which image class labels are used to derive bounding box annotations. It is known that WSOL methods have drawbacks of overly relying on inter-class discriminative features and failing to generalize to classes unseen during the training phase.

We argue that intra-class similarity is a more natural objective for the problem of localizing objects belonging to a target class. "Query object localization" emphasizes more about the class-specific nature of a localization task, with the relevance defined by the similarity to query images. A similar problem is image co-localization (Tang et al., 2014; Wei et al., 2017), in which the task is to identify the common objects within a set of images. Co-localization approaches exploit the common characteristics across images to localize objects. Being unsupervised, co-localization approaches could suffer from ambiguity if there exist multiple common objects or parts, e.g., bird head and body (Jaderberg et al., 2015), which may provide unwanted common objects as output.

### 2.2 POLICY ADAPTATION

There seems to be a contradiction between the goals of training an agent with high task-specificity and better generalization performance to new situations at the same time. The key to reconcile these two goals lies in the usage of a small set of examples. There has been a paradigm shift from training static models defined with parameters to dynamic ones defined together with a support set (Vinyals et al., 2016; Snell et al., 2017), proved to be very effective in few-shot classification.

Besides the efforts of employing meta learning to adjust models, fine-tuning on a pre-trained model (Oquab et al., 2014; Finn et al., 2017; Julian et al., 2020) has also been widely used in transferring knowledge from data-abundant to data-scarce tasks. When reward signal is not available, (Hansen

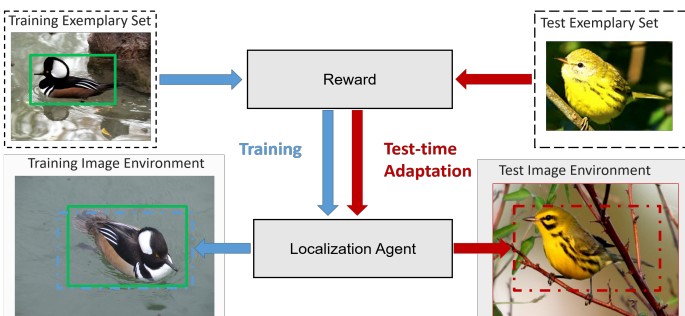

Figure 1: RL-based query object localization: training and adaptation. Image-box pairs are only available in the training phase. Test environments include images without box annotations and an exemplary set specifying a different task objective. Thus, the trained policy needs to be adapted.

et al., 2021) proposed a policy adaptation approach, in which the intermediate representation is fine-tuned via optimizing a self-supervised auxiliary loss while the controller is kept fixed. Our work shares the same motivation of test-time training (Sun et al., 2020), but we focus instead on the settings where the controller needs to be adapted or even repurposed for new tasks.

# 3 LEARNING TRANSFERABLE REWARD FOR LOCALIZING QUERY OBJECTS

## 3.1 RL FORMULATION FOR QUERY OBJECT LOCALIZATION

Given a set of images $\mathcal{I}$ with known class labels, the task of query object localization is to find the location $b$ of the bounding box most relevant to the query object, specified in a small set of exemplary images $\mathcal{E}$. Following the Markov Decision Process (MDP) framework for object localization (Caicedo & Lazebnik, 2015; Jie et al., 2016), this task can be formulated as a RL problem. In the training phase, a localization agent learns to take actions of moving the bounding box to maximize a reward function reflecting localization accuracy. In the test phase, the agent is informed by a small test exemplary set about a new task objective. Figure 1 shows an overview of the query object localization problem and the RL formulation, with components listed as follows.

- **Environment:** Raw pixels of a single image $I_i$, without candidate proposal boxes.
- **Action:** Discrete actions facilitating a top-down search, including five scaling: scale towards top left/right, bottom left/right, center; eight translation transformations: move left/right/up/-down, enlarge width/height, reduce width/height; plus one stay action.
- **State:** Pooled features from the current box and an internal state within RNN that encodes information from history observations.
- **Reward:** Improvements the agent made in localizing the query images (in different tasks).

## 3.2 PRE-TRAINING ORDINAL REWARD SIGNAL

For training an agent to localize different objects according to queries, the reward function needs to be transferable. Existing deep RL approaches for object localization (Caicedo & Lazebnik, 2015; Jie et al., 2016) use its ground-truth object bounding box $g_i$ as the reward signal,

$$R = \text{sign}(\text{IoU}(b_t, g_i) - \text{IoU}(b_{t-1}, g_i)), \tag{1}$$

where $\text{IoU}(b_t, g_i)$ denotes the Intersection-over-Union (IoU) between the current window $b_t$ and the corresponding ground-truth box $g_i$[2], and $\text{IoU}(b, g) = \text{area}(b \cap g)/\text{area}(b \cup g)$. Similar to the bounding box regression approaches which learn a mapping $\phi : I \mapsto g$, the image and box must be paired. However, annotated image-box pairs $(I, g)$ may be scarce in both the training and testing phases. The reward signal defined in Eq. 1 is instance-wise, which is not transferable across images.

To address this problem, a natural idea is to define the reward signal based on learned representations of the cropped images by current window $b_t$ and the ground-truth window $g$. Given their $M$-dimensional representations $\mathbf{b_t}$ and $\mathbf{g}$ produced by an embedding function $f : \mathbb{R}^D \mapsto \mathbb{R}^M$ from $D$- dimensional image feature vectors, a distance function $d : \mathbb{R}^M \times \mathbb{R}^M \mapsto [0, +\infty)$ returns the embedding distance $d(\mathbf{b_t}, \mathbf{g})$. However, an embedding distance based on off-the-shelf pre-trained networks is not adequate, since it may not decrease monotonically as the agent approaches to the ground-truth box $g$. As a result, the embedding distance based reward signal may be less effective than Eq. 1 (as shown in Section 4). Therefore, an embedding function customized for the downstream RL-based localization task is needed.

**Ordinal metric learning.** We propose to use an ordinal embedding based reward signal. For any two perturbed boxes $b_j, b_k$ from ground truth $g$, embeddings $\mathbf{b_j}, \mathbf{b_k}, \mathbf{g}$ are learned, such that the relative preferences between any pair of boxes are preserved in the Euclidean space,

$$\rho_j > \rho_k \Leftrightarrow ||\mathbf{b_j} - \mathbf{g}|| < ||\mathbf{b_k} - \mathbf{g}||, \quad \forall j, k \in \mathcal{C}, \tag{2}$$

where $\rho_j$ and $\rho_k$ denote the preference (derived from either IoU to ground-truth box or ordinal feedback from user), and $\mathcal{C}$ is an ordinal constraint set constructed by sampling box pairs around

---

[2]Bounding box are denoted by lowercase lightface letters, e.g., $g$, while embedding vectors are denoted by boldface letters, e.g., $\mathbf{g} \in \mathbb{R}^d$ is a $d$-dimensional vector.

$g$. This problem is originally posed as non-metric multidimensional scaling (Agarwal et al., 2007; Jamieson & Nowak, 2011). Although we apply a very simple pairwise-based approach, there exist other extensions such as the listwise-based (Cao et al., 2007), quadruplet-based (Terada & Luxburg, 2014) and landmark-based (Anderton & Aslam, 2019; Ghosh et al., 2019) approaches.

**Loss function.** In this paper, we define preference $\rho$ as the IoU of box $b$ to the ground-truth box $g$, i.e., $\rho = \text{IoU}(b, g)$. We choose to optimize a triplet loss (Hermans et al., 2017) for learning the desired embeddings,

$$\mathcal{L}_{\text{triplet}} = \sum_{\text{anchor, ordinal pairs}} \max\big(m + d(\mathbf{a}, \mathbf{p}) - d(\mathbf{a}, \mathbf{n}), 0\big), \tag{3}$$

where $\mathbf{a}$ is the "anchor" embedding that will be discussed in detail later, $\mathbf{p}, \mathbf{n}$ are the "positive" and "negative" embeddings for boxes with larger and smaller IoUs with ground truth box $g$, respectively, $m$ is a margin hyper-parameter. We learn an embedding space consistent with local ordinal constraints on positive and negative data obtained via data augmentation such as box perturbation.

**Box Perturbation** Assuming the training exemplary set $\mathcal{E}_{\text{train}}$ contains both image $I$ and box $g$, we adopt a tailored data augmentation scheme - *box perturbation*, in which a pair of boxes are randomly sampled from $I$ with different IoUs to $g$. We compare the efficiency of sampling schemes on generating augmented bounding box pairs, (a) *Random sampling*, where the pair of boxes are generated completely randomly, and (b) *Group-based sampling*, where dense boxes with variant scales are first generated, and then divided into $M = 10$ groups according to the IoU with ground-truth box. Each group has an IoU interval of $1/M$. The sampling is first done on the group level, i.e., 2 out of $M$, then sample one box from each group. Thus, the sampled boxes are likely to cover more diverse cases comparing to random sampling. We have found that using IoU-based partition scheme is more effective than random sampling.

**The choice of anchor.** The anchor embedding $\mathbf{a}$ in Eq. 3 is not restricted to the cropped image from the same image. For example, it could be replaced by the prototype embedding (Snell et al., 2017) of the exemplary set $\mathcal{E}$, $\mathbf{c} = 1/|\mathcal{E}| \sum_{i \in \mathcal{E}} \mathbf{g}_i$, where $\mathbf{g}_i$ is the embedding of the cropped image $I_i$ by ground-truth box $g_i$. If images from multiple classes are of interest, the prototype can be further made to be class-dependent, or clustering-based. A prototype as the anchor enables test-time policy adaptation without image-box pairs in the exemplary set $\mathcal{E}_{\text{test}}$. In comparison, supervised methods such as Faster RCNN would require image-box pairs for fine-tuning on new tasks. In some experiments (Appendix B.1), we find that using prototype-based embedding as the anchor could lead to better generalization performance than using $\mathbf{g}$. It is also interesting to explore other treatment on the exemplary set, such as permutation-invariant architectures (Zaheer et al., 2017; Ilse et al., 2018; Lee et al., 2019).

### 3.3 LOCALIZATION AGENT TRAINING

As discussed in Section 3.2, we use the ordinal embedding rather than the bounding box coordinates to quantify the improvement that an agent makes, and the continuous-valued reward for the agent

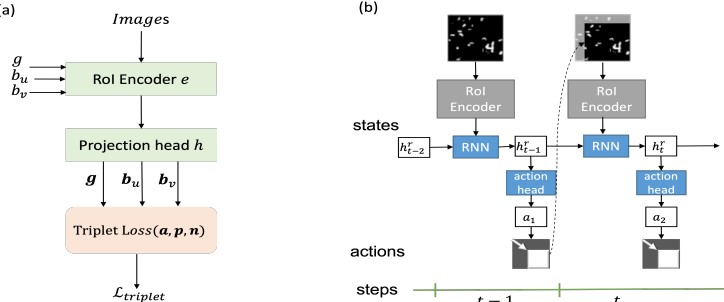

Figure 2: (a) Modules for ordinal representation learning. The RoI encoder extracts RoI feature that will be used as the state representation for localization. The projection head learns ordinal embedding $\mathbf{b}$ for computing reward. (b) Controller network. In each step, the agent takes the output from the RoI encoder as state, while it also maintains an internal hidden state $h^r$ within a RNN, which encodes information from history observations. Then it outputs the action for next step.

moving from state $s'$ to $s$ takes the following form,

$$R(s, s') = ||\mathbf{b}_{t-1} - \mathbf{c}|| - ||\mathbf{b}_t - \mathbf{c}||, \tag{4}$$

where $\mathbf{c}$ is the prototype embedding, $\mathbf{b}_t$ is the embedding of the cropped region at step $t$. We expect positive reward if the embedding distance between cropped image $\mathbf{b}_t$ and query image $\mathbf{a}$ is decreased.

**State representation.** Note that a good representation for defining reward may not necessarily be a good state representation at the same time - it may not contain enough information guiding the agent taking the right actions. Chen et al. (2020) suggests that adding a projection head between the representation and the contrastive loss substantially improves the quality of the learned representation. We find that the use of projection head is crucial in balancing the two objectives in our task. The network architecture is shown in Figure 2(a), in which an MLP projection head is attached after an (Region of Interest) RoI encoder. The ROIAlign module handles boxes of different sizes.

**Policy network.** Starting from a whole image as input, the agent is trained to select actions to transform the current box at each step, maximizing the total discounted reward. We use policy gradient based on a recurrent neural network (RNN) (Mnih et al., 2014) with a vector of history actions and states. More detailed architecture can be found in Figure 2(b) and Appendix Figure 6. For all experiments, we apply policy gradient (REINFORCE) (Williams, 1992) with entropy loss regularization to encourage exploration. The main algorithm is outlined in Algorithm 1.

---

**Algorithm 1:** Training localization agent using the proposed ordinal reward signal.

---
**Require :** initial policy $\pi_\theta$, batch size $N$, pre-trained RoI encoder $e$ and projection head $h$, learning rate $\alpha$
**Output :** policy network $\pi_\theta$
**for** *sampled minibatch* $\{x_k\}_{k=1}^N$ **do**
    **for** *k=1,...N* **do**
        Construct an exemplary set $\mathcal{E}_{\text{batch}}$
        $\mathbf{g_j} \leftarrow h(e(x_k, g_j))$ `// compute the ordinal embedding in Figure 2(a)`
        $\mathbf{c}_k \leftarrow \frac{1}{|\mathcal{E}_{\text{batch}}|} \sum_{j \in \mathcal{E}_{\text{batch}}} \mathbf{g}_j$ `// compute the prototype vector`
        $\tau_k \leftarrow \{ s_1, a_1, ...s_T, a_T \}$ by running policy $\pi_\theta$ `// sample trajectory`
        $\mathbf{b_t} \leftarrow$ take action $a_t$, and compute embedding $h(e(x_k, b_t))$ for current window $b_t$
        $R_k^t \leftarrow ||\mathbf{b}_{t-1} - \mathbf{c}_k|| - ||\mathbf{b}_t - \mathbf{c}_k||$ `// compute reward in equation 4`
    **end**
    $\theta \leftarrow \theta + \alpha \nabla_\theta \sum_{k=1}^N \left[ \log \pi_\theta \left( \sum_{t'=t}^T \gamma^{t'-t} R_k^t \right) - \lambda \cdot \pi_\theta \log \pi_\theta \right]$ `// policy update`
**end**

---

### 3.4 TEST-TIME ADAPTATION

During test time, the agent has the option of further updating the policy network using the received reward from Eq. 4. To match test conditions, the training batch is split into two groups, and $\mathbf{c}$ is computed on a small subset that does not overlap with the training images to localize; During test-time adaptation, $\mathbf{c}$ becomes the prototype of the test exemplary set $\mathcal{E}_{\text{test}}$. Different from $\mathcal{E}_{\text{train}}$, only cropped images are needed in $\mathcal{E}_{\text{test}}$ as shown in Figure 1.

## 4 EXPERIMENTAL RESULTS

In this section, we evaluate the generalization ability of the ordinal embedding as well as the performance of the trained localization agent on both source and target domains. The embedding-based reward not only improves the RL training, but also enables test-time adaptation of the learned policy. Experimental results demonstrate the effectiveness of our approach, with empirical comparisons to fine-tuning, co-localization, few-shot and Faster RCNN object detection baselines.

**Implementation details.** To evaluate the learned ordinal embedding, we use *OrdAcc* defined as the percentage of images within which the preference between a pair of perturbed boxes is correctly predicted. To evaluate object localization performance, we use the Correct Localization (*CorLoc*) (Deselaers et al., 2012) metric, which is defined as the percentage of images correctly localized according to the criterion $\text{IoU}(b_p, g) \geq 0.5$, where $b_p$ is the predicted box and $g$ is the ground-truth

box. Mean and standard deviation (displayed as subscript) are reported from 10 independent runs. We evaluate our approach on distorted versions of the MNIST handwriting, the CUB-200-2011 birds (Wah et al., 2011), and the COCO (Lin et al., 2014) dataset. For the MNIST dataset, we use three convolutional layers with ReLU activation after each layer as the image encoder. For the CUB and the COCO datasets, we adopt layers before *conv5_3* of VGG-16 pre-trained on ImageNet as the encoder, unless otherwise specified. More details are presented in Appendix A.2.

## 4.1 POLICY ADAPTATION DURING TEST TIME

We demonstrate the performance improvement with test-time policy adaptation. Through all the experiments, we assume source domain contains abundant data annotations, and target domain annotations are only available in an exemplary set of size 5. We compare our policy adaptation scheme with a standard fine-tuning scheme on the pre-trained policy network.

**Results on the corrupted MNIST dataset.** For the new class adaptation experiment, we use 50 "digit 4 images under random patch background noises" to train the ordinal embedding and the localization agent. The results on policy adaptation to *new digits* (other than 4) are shown in Table 1. Row 1 illustrates the transferability of the ordinal embedding reward, trained prototype embedding of a subgroup without the training instance, and evaluated using instance embedding from the same test image ("$OrdAcc$"). Rows 2 to 4 list the resulting localization accuracy after direct generalization ("before"), fine-tuning on the exemplary set ("fine-tune"), and adaptation using all test images ("adapt"), respectively. Our policy adaptation approach produces a substantial improvement over direct generalization, while fine-tuning approach experiences overfitting on the limited exemplary set. For the background adaptation experiment, we train on 50 digit-3 images under random patch noise, and test on digit-2 images under all four noises. The localization accuracy on both source and *new backgrounds* environment are shown in Table 2, significant improvements are achieved using our policy adaptation scheme.

Table 1: $OrdAcc$ (%) and $CorLoc$ (%) on new digits environment.

|  | 0 | 1 | 2 | 3 | 5 | 6 | 7 | 8 | 9 | mean |
|---|---|---|---|---|---|---|---|---|---|---|
| $OrdAcc$ | $91.9_{0.7}$ | $90.3_{2.9}$ | $92.1_{1.8}$ | $92.0_{0.2}$ | $92.7_{0.6}$ | $92.6_{0.2}$ | $90.7_{0.7}$ | $92.0_{0.7}$ | $90.5_{0.7}$ | $91.6_{0.5}$ |
| before | $94.2_{0.6}$ | $84.1_{1.5}$ | $88.7_{1.7}$ | $86.5_{1.8}$ | $81.2_{1.4}$ | $91.9_{0.3}$ | $89.5_{0.7}$ | $93.0_{1.0}$ | $90.8_{0.4}$ | $88.9_{1.0}$ |
| fine-tune | $93.3_{2.5}$ | $80.4_{3.9}$ | $84.5_{3.5}$ | $84.9_{1.5}$ | $78.8_{2.7}$ | $87.3_{2.9}$ | $82.2_{5.4}$ | $87.7_{4.5}$ | $85.6_{5.3}$ | $83.7_{4.5}$ |
| adapt | $99.8_{0.2}$ | $95.6_{0.6}$ | $98.1_{0.4}$ | $97.9_{0.4}$ | $88.3_{0.5}$ | $99.1_{0.2}$ | $98.6_{0.9}$ | $99.8_{0.3}$ | $99.2_{0.4}$ | $97.4_{0.4}$ |

Table 2: $CorLoc$ (%) when adapted to other background on corrupted MNIST dataset.

| adapt | random patch | clutter | impulse noise | gaussian noise | mean |
|---|---|---|---|---|---|
|  | $97.6_{0.4}$ | $39.6_{0.5}$ | $22.1_{0.7}$ | $66.2_{2.0}$ | $56.4$ |
| ✓ | $\mathbf{100.0_{0.0}}$ | $\mathbf{97.4_{0.3}}$ | $\mathbf{99.9_{0.1}}$ | $\mathbf{100.0_{0.0}}$ | $\mathbf{99.3}$ |

**Results on the CUB dataset.** We also evaluate the policy adaptation performance on the CUB dataset. The localization agent is trained on 15 species from the "Warbler" class, and adapted to different classes of "Warbler" (5 new species), "Wren", "Sparrow", "Oriole", "Kingfisher", "Vireo", and "Gull". Each test class contains a single bird class. We also implement deep descriptor transforming (DDT) (Wei et al., 2017), a deep learning based co-localization approach, and add it to the comparison.

Table 3: $CorLoc$ (%) when adapted to other species/classes on CUB dataset.

|  | adapt | warbler (new) | wren | sparrow | oriole | kingfisher | vireo | gull | mean |
|---|---|---|---|---|---|---|---|---|---|
| DDT |  | $73.8$ | $78.6$ | $71.2$ | $74.5$ | $78.0$ | $69.2$ | $93.3$ | $76.9$ |
| Ours |  | $85.5_{1.1}$ | $82.9_{2.6}$ | $81.3_{3.7}$ | $77.9_{0.7}$ | $78.9_{0.6}$ | $82.2_{4.6}$ | $86.3_{3.5}$ | $82.1$ |
|  | ✓ | $\mathbf{89.7_{1.1}}$ | $\mathbf{91.0_{1.1}}$ | $\mathbf{89.3_{1.8}}$ | $\mathbf{85.0_{0.8}}$ | $\mathbf{85.9_{4.4}}$ | $\mathbf{90.0_{0.9}}$ | $\mathbf{93.9_{0.5}}$ | $\mathbf{89.3}$ |

**Results on the COCO dataset.** Given a target domain that one would like to deploy models to, a natural question is whether one should collect labeled data from an abundant number of source domains or from one specific class with potential lower data collection cost. On the COCO dataset, we train on one of the five classes: *cat, cow, dog, horse, zebra*, then adapt to another four classes. The results are shown in Figure 3. The models perform best using ordinal embedding reward with

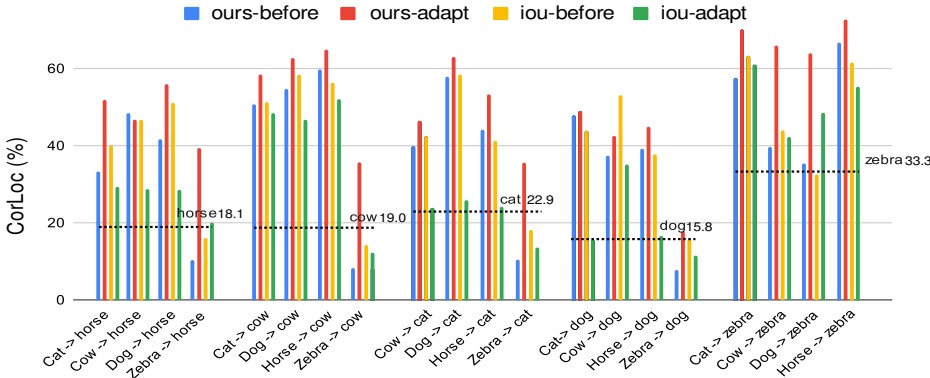

Figure 3: *CorLoc* (%) Comparison with IoU-based reward on COCO dataset. "ours": ordinal embedding based reward; "iou": IoU based reward, ImageNet pre-trained embedding. Dotted lines are the results of directly training on target class, using embedding based reward but from a ImageNet pre-trained model, indicating the advantage of our approach, with ordinal embedding learned from a single source class and policy adaptation.

adaptation (see "ours-adapt"). It shows that, although an agent is trained in a highly specialized way on only one-source tasks, it can still be flexibly generalized to different target tasks. One interesting observation is that, it is easy to transfer from other four classes to zebra, but not vice versa. A possible explanation might be that, the embedding net is biased on textures, and texture information is less adaptable than shape information.

## 4.2 COMPARISON TO FEW-SHOT OBJECT DETECTORS

Few-shot object detection (Kang et al., 2019; Yan et al., 2019; Wang et al., 2019; 2020) methods are similar to our work in adapting models to a new test case with the help of limited annotations, but they have a different requirement on the number of training classes. Being different in the generalization mechanism, they usually require multiple classes in both stages of training and fine-tuning. On the COCO dataset, we compared the performance of our methods with TFA (Wang et al., 2020), and Meta-RCNN (Yan et al., 2019), on the same *one-way 5-shot* test setting. TFA adopts two-stage finetuning of object detector, while Meta-RCNN incorporates additional meta-learner to acquire class-level meta knowledge for generalization to novel classes. These two few-shot baselines are trained on 60 base classes and fine-tuned on all 80 classes, while our model is trained from one single class of the 5 classes randomly selected from the base classes set: *elephant, sheep, giraffe, laptop* and *suitcase*. Table 4 shows that with ordinal embedding, our agent achieves better performance even without adaptation, and the performance can be further improved after adaptation.

Table 4: *CorLoc* (%) Adaptation comparison with other methods on COCO dataset. "TFA w/ fc": TFA with a cosine similarity based box classifier; "TFA w/ cos": TFA with a normal FC-based classifier; "FRCN+ft-full": Meta-RCNN with Faster RCNN as detector and finetuning both the feature extractor and the box predictor.

| target | cat | dog | cow | horse | mean |
|---|---|---|---|---|---|
| FRCN+ft-full (Yan et al., 2019) | 13.1 | 3.1 | 3.7 | 7.4 | 6.8 |
| TFA w/ fc (Wang et al., 2020) | 29.1 | 13.0 | 5.0 | 10.7 | 14.4 |
| TFA w/ cos (Wang et al., 2020) | 28.0 | 10.3 | 4.5 | 8.9 | 12.9 |
| Ours-before | 23.0 | 20.6 | 24.5 | 21.2 | 22.3 |
| Ours-adapt | 40.3 | 33.5 | 43.1 | 40.2 | 39.3 |

## 4.3 COMPARE WITH SUPERVISED BASELINE - FASTER RCNN

We compare our framework with a strong supervised object localization baseline, Faster RCNN (Ren et al., 2016). Both methods are trained in one class (foreground *vs.* background as the classification labels in Faster RCNN) and adapted to a different class. We fine-tune the pre-trained VGG-16 model and test on each of the five classes: *cow, cat, dog, horse, zebra*. The results on source domain are

shown in Table 5. It shows that Faster RCNN can also be made into a class-specific model and it still yields superior performance on source domain. On the target domain, we fine-tune Faster RCNN using only query set for each target class. The results are shown in Table 6. It can be seen that our method works better on new classes with test-time adaptation over the traditional fine-tuning of Faster RCNN. Note that fine-tuning requires image-box pairs in the target domain while our policy adaptation approach does not.

Table 5: $CorLoc$ (%) comparison with Faster RCNN on source domain.

| method | cow | cat | dog | horse | zebra |
|---|---|---|---|---|---|
| Faster RCNN (Ren et al., 2016) | 70.37 | 89.82 | 85.81 | 92.65 | 85.71 |
| ours | 70.37 | 68.46 | 61.26 | 61.28 | 79.36 |

Table 6: $CorLoc$ (%) comparison with Faster RCNN fine-tuned on target domain.

| | before fine-tune | before adapt | after fine-tune | after adapt |
|---|---|---|---|---|
| | Faster RCNN | ours | Faster RCNN | ours |
| cat ->horse | 20.93 | **33.32** | 37.73 | **51.89** |
| cow ->horse | **54.79** | 48.41 | **68.04** | 46.80 |
| cog ->horse | 38.52 | **41.50** | **58.01** | 55.89 |
| zebra ->horse | 1.12 | **10.29** | 6.04 | **39.22** |
| cat ->cow | 40.52 | **50.85** | 58.55 | **58.58** |
| dog ->cow | 54.55 | **54.63** | **70.11** | 62.86 |
| horse ->cow | **72.11** | 59.52 | **75.35** | 64.83 |
| zebra ->cow | 1.23 | **8.14** | 5.86 | **35.56** |
| cow ->cat | **46.12** | 39.84 | **53.85** | 46.42 |
| dog ->cat | **67.62** | 57.97 | **77.07** | 63.12 |
| horse ->cat | 36.67 | **44.25** | 36.67 | **53.39** |
| zebra ->cat | 0.55 | **10.45** | 4.09 | **35.73** |
| cat->dog | **58.98** | 47.81 | **66.50** | 48.94 |
| cow ->dog | **45.85** | 37.28 | **51.69** | 42.33 |
| horse ->dog | **41.04** | 39.07 | **47.69** | 44.77 |
| zebra ->dog | 0.68 | **7.74** | 3.4 | **17.73** |
| cat ->zebra | 10.64 | **57.58** | 37.97 | **70.28** |
| cow ->zebra | 4.42 | **39.64** | 19.64 | **65.80** |
| dog ->zebra | 2.29 | **35.27** | 15.88 | **63.91** |
| horse ->zebra | 7.86 | **66.82** | 29.3 | **72.83** |
| mean | 30.32±25.0 | **39.51±17.9** | 41.17±25.3 | **52.04±13.9** |

## 4.4 ABLATION STUDIES

We analyze the effectiveness of ordinal embedding and RL component separately on COCO dataset (Appendix B.1 provides more in-depth analysis on the corrupted MNIST dataset, including policy gradient *vs.* deep Q-Network, continuous *vs.* binary reward). First, we remove RL, and substitute it with a simple linear search approach. Specifically, we adopt selective search (Uijlings et al., 2013) for candidate boxes generation. The candidate boxes are ranked according to their embedding distances to the prototype, and the one with the smallest embedding distance is returned as the final output. We consider two backbone networks as the embedding, including ImageNet pre-trained VGG-16 and Faster RCNN VGG-16 trained locally on COCO dataset. We also compare both with and without the ordinal component, making it a $2 \times 2$ ablation study. It can be seen from the blue and green bars from Figure 4 and Appendix Figure 10 that with ordinal structure, the ranking method performs much better. We find that pre-training with the proposed ordinal loss significantly improves the rank consistency of these backbone networks (Appendix B.1).

In contrast, being able to learn across different images, the RL localization agent shows more tolerance to the defects of the reward function. We analyze the benefits of RL component. From both Figure 4 and Figure 10, when using ordinal embedding, the model with RL is generally better than ranking, especially after adaptation.

Ordinal component is necessary for several possible reasons. First, the ordinal component makes the generic pre-trained network more specific to the downstream object localization task. Second, it

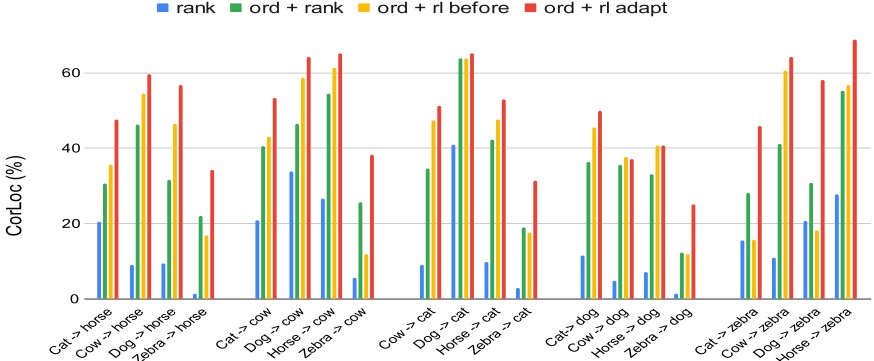

Figure 4: $CorLoc$ (%) comparison with ranking method using Faster RCNN pre-trained backbone.

helps remove the dataset bias if the pre-trained network is non-native. Third, ordinal pre-training encourages the reward signal to reflect step-by-step improvements.

Encouraged by the strong baseline performance of Faster RCNN on both source and target domain (detailed in Table 5 and Table 6), we investigate whether it is adequate to use Faster RCNN embedding in the RL reward function, with or without ordinal pre-training. Since Faster RCNN emebedding is also trained for object localization on the COCO dataset, it can separate out the effects of the ordinal pre-training component better. The result are shown in Table 7. It can be seen that without ordinal pre-training, the performance degrades significantly and the $CorLoc$ is much lower even in source domain. More results of different backbones are presented in Appendix Table 15 and Table 16, from which the test-time adaptation still brings a large margin of improvement.

Table 7: $CorLoc$ (%) RL with Faster RCNN embedding w/ and w/o ordinal pre-training on source domain.

| method | cow | cat | dog | horse | zebra |
|---|---|---|---|---|---|
| Faster RCNN backbone | 25.93 | 13.17 | 12.16 | 16.18 | 28.57 |
| Faster RCNN backbone + Ord | **66.67** | **72.46** | **61.59** | **60.29** | **71.25** |

## 5   CONCLUSION AND FUTURE WORK

We propose an ordinal representation learning based reward, for training a localization agent to search queried objects of interest. In particular, we use a small exemplary set as a guidance signal for delivering learning objectives without ambiguity. Meanwhile, we use test image environments to inform the agent about the domain shifts, without requiring image-box pairs during test time.

Instead of jointly training a localization and classification model, we learn box annotations from image class labels, in a similar spirit to weakly-supervised learning. Empirically, we focus extensively on the *transfer learning* setting from one single data abundant training task to data-scarce (one-way few-shot) test tasks (Taylor & Stone, 2009; Zhu et al., 2020). To make a trained RL agent generalizable to new unseen test cases, one can either expose the agent to as many cases as possible during training, or make the agent more specific but adaptable upon changed scenarios. Our approach belongs to the latter.

The transferability of our reward signal from training to test crucially relies on the generalization ability of the learned ordinal representation. If the ordinal preference does not hold in the test domain, the proposed test-time policy adaptation scheme will not work. By adapting the representation with self-supervised objectives (Hansen et al., 2021), this issue might be remedied.

Although our work focuses on the generalization ability of a single object localizer, extensions to multiple objects can possibly be done by applying it multiple times and marking the found regions, in a similar manner as (Caicedo & Lazebnik, 2015). Curriculum learning with a designed sequence of targets in the exemplary set may guide the agent toward solving challenging localization tasks. We leave those extensions for future study.

## ACKNOWLEDGEMENT

This research has been partially funded by the following grants ARO MURI 805491, NSF IIS-1793883, NSF CNS-1747778, NSF IIS 1763523, DOD-ARO ACC-W911NF, NSF OIA-2040638 to D. Metaxas. We would like to thank anonymous reviewers for their comments and suggestions.

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

# A  APPENDIX: IMPLEMENTATION DETAILS

## A.1  ALGORITHM PIPELINE

General supervised training methods are usually class-agnostic and require exposure to a large number of training classes, box-image pairs, and foreground and background variations in order to generalize well. In contrast, we allow specialized agent to be trained, with the ability of adaptation to changes during the test time. Our approach is based on the feature similarity with query images, which departs from previous bounding-box regression and RL approaches based on objectiveness. Compared to general supervised training or fine-tuning methods, our approach is able to flexibly make use of various types of data in these phases. This is summarized in Table 8.

Table 8: Breaking up the requirements on data and labels in training and adaptation

|  | Supervised methods | | Our approach | | |
| --- | --- | --- | --- | --- | --- |
|  | Training | Fine-tuning | Ordinal embedding | Agent training | Test-time adaptation |
| Image-box pairs | ✓ | ✓ | ✓ | ✗ | ✗ |
| Unlabeled images | ✗ | ✗ | ✗ | ✓ | ✓ |
| Exemplar images | ✗ | ✗ | ✓ | ✓ | ✓ |

Our agent can both take human feedback in terms of the exemplary set and perform test-time policy adaptation using unlabeled test data. It includes three stages: ordinal representation pre-training, RL agent training, and test-time adaptation. Details are as follows:

**Stage 1: Pre-train ordinal embedding for state representation and reward**  In this stage, we assume pairs of ground-truth bounding box and training image are available. We train the ordinal embedding by attaching a projection head after RoI encoder, as Figure 5 shows. RoI encoder is composed of image encoder and a RoIAlign layer (He et al., 2017). It extracts corresponding bounding box feature directly from image feature output by image encoder. After training, all the modules are fixed. The output of RoI Encoder then becomes the state for agent, and output of projection head is then used for reward computation.

On Corrupted MNIST (cMNIST) dataset, we attach a decoder in order to train the encoder, but discard it during inference. The loss function is defined as

$$\mathcal{L} = \mathcal{L}_{reconstruct} + \lambda_1 \cdot \mathcal{L}_{triplet}, \tag{5}$$

where we set $\lambda_1 = 0.1$, and $\mathcal{L}_{reconstruct}$ is mean square error loss to measure the reconstruction ability of the autoencoder. On CUB and COCO dataset, we adopt the pre-trained ImageNet encoder, thus no additional decoder is needed. The triplet loss $\mathcal{L}_{triplet}$ is optimized, such that the embedding of positive bounding boxes are closer to the embedding of anchor than negative boxes. Different from traditional definition of triplet loss, our positive and negative boxes are not fixed. When comparing two perturbed boxes, the positive box is always the one with larger IoU. Thus, a box with IoU=0.2 could be positive box if the negative box has even smaller IoU. The anchor box is not restricted to ground truth box of the image instance. For example, we use the mean vector of ordinal embeddings in this paper, computed from ground-truth boxes of images from the exemplary set.

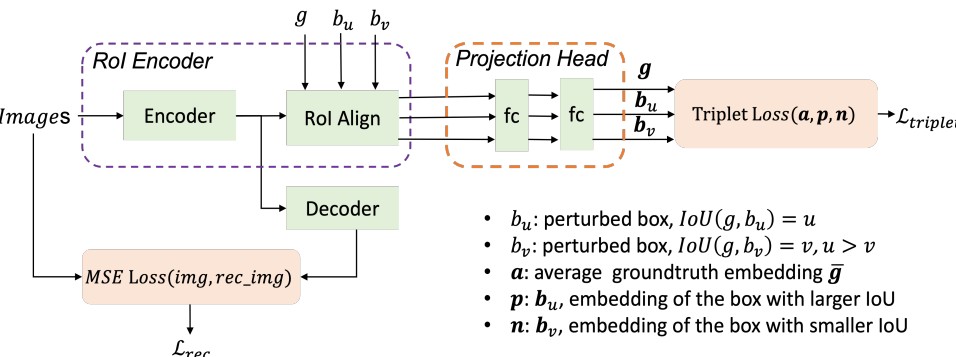

Figure 5: Learning RoI Encoder and Projection Head.

Table 9: Network and loss updating details for different stages of the method

| | Configuration | | | Training | | Testing of the RL agent | |
|---|---|---|---|---|---|---|---|
| Modules | Objective | Network | Exemplary set | Ordinal pre-training | Policy training | before adaptation | after adaptation |
| ROI Encoder | NA* | VGG-16/ViT | $\mathcal{E}_{\text{train}}$ | Frozen | Frozen | Frozen | Frozen |
| Projection Head | Ordinal loss $\mathcal{L}_{\text{triplet}}$ | MLP | $\mathcal{E}_{\text{train}}$ | Train | Frozen | Frozen | Frozen |
| Controller | Reward | RNN | $\mathcal{E}_{\text{test}}$ | NA | Train | Frozen | Updated |

\* For cMNIST dataset, ROI encoder is trained under loss Eq.5. For other datasets, we directly load the off-the-shelf pre-trained network.

**Stage 2: Localization agent training** In this stage, we assume the ground-truth bounding box is available for each training image. All the models are trained and tested on the same datasets as stage 1. We fix all the layers in RoI encoder and projection head. The adopted RNN model is shown in Figure 6. In each step, the agent takes the RoI encoder output $e_{\text{roi}}$ and the hidden state of RNN $h_{t-1}$, concatenated as the state representation. Then it predicts the action for next step by sampling according to the logits. The loss is defined as

$$\mathcal{L}_{\text{agent}} = \mathcal{L}_{\text{policy}} + \lambda_2 \cdot \mathcal{L}_{\text{entropy}},$$
$$\mathcal{L}_{\text{entropy}} = -\pi \log \pi, \quad \mathcal{L}_{\text{policy}} = -(R - \bar{R}) \cdot \log \pi \tag{6}$$

where $\mathcal{L}_{\text{entropy}}$ prevents the agent from being stuck in the local minimum, $\pi$ is the policy, and $R$ is the reward defined in Eq. 4.

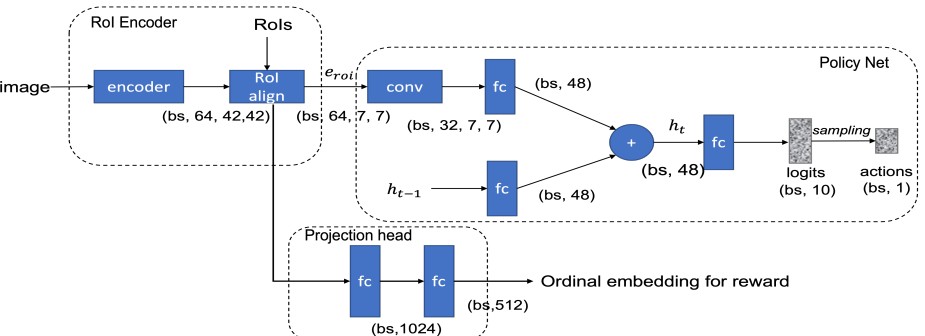

Figure 6: Policy network architecture.

**Stage 3: Test-time adaptation** To adapt to new class, we make use of a limited number of groundtruth-box cropped images in the test exemplary set. We leverage abundant unlabelled images and this exemplary set to update the agent. We fix the RoI encoder and projection head, and only update the agent. The process is almost the same as stage 2, except that **c** in reward (Eq. 4) is the prototype of the test exemplary set.

In summary, the configuration of different model components and how they are updated is listed in Table 9. The losses for each dataset and training stage are in Table 10.

Table 10: Summary of losses used on different datasets.

| | dataset | loss |
|---|---|---|
| | cMNIST | $\mathcal{L} = \mathcal{L}_{reconstruct} + \lambda_1 \cdot \mathcal{L}_{triplet}, \lambda_1 = 0.1$ |
| stage 1 | CUB | $\mathcal{L} = L_{triplet}$ |
| | COCO | $\mathcal{L} = L_{triplet}$ |
| | cMNIST | $\mathcal{L}_{agent} = \mathcal{L}_{policy} + \lambda_2 \cdot \mathcal{L}_{entropy}, \lambda_2 = 6$ |
| stage 2 | CUB | $\mathcal{L}_{agent} = \mathcal{L}_{policy} + \lambda_2 \cdot \mathcal{L}_{entropy}, \lambda_2 = 0.5$ |
| | COCO | $\mathcal{L}_{agent} = \mathcal{L}_{policy} + \lambda_2 \cdot \mathcal{L}_{entropy}, \lambda_2 = 0.5$ |
| | cMNIST | $\mathcal{L}_{agent} = \mathcal{L}_{policy} + \lambda_2 \cdot \mathcal{L}_{entropy}, \lambda_2 = 0.5$ |
| stage 3 | CUB | $\mathcal{L}_{agent} = \mathcal{L}_{policy} + \lambda_2 \cdot \mathcal{L}_{entropy}, \lambda_2 = 0.5$ |
| | COCO | $\mathcal{L}_{agent} = \mathcal{L}_{policy} + \lambda_2 \cdot \mathcal{L}_{entropy}, \lambda_2 = 0.5$ |

## A.2 Experiments Details

For MNIST, we use three convolutional layers with ReLU activation after each layer as image encoder, while the same but mirrored structure as decoder to learn an autoencoder, and then attach ROIAlign layer followed by two fully connected (*fc*) layers as projection head for ordinal reward learning. For the CUB and the COCO datasets, we adopt layers before *conv5_3* of VGG-16 pre-trained on ImageNet as encoder unless otherwise specified. The projection head uses the same structure as before but with more units for each *fc* layer. All of our models were trained with the Adam optimizer (Kingma & Ba, 2015). We set margin $m = 60$ in all the experiments heuristically. All the models take less than one hour to finish training, implemented on PyTorch on a single NVIDIA A100 GPU.

We list experiment details for the datasets used in Section 4 as follows.

**Corrupted MNIST Dataset.**  Examples of the four types of corrupted MNIST (cMNIST) images are shown in Figure 7. In stage 1 and 2, we randomly sample 50 images of one class in original MNIST training set, and add noise as our training set. During testing, the models are evaluated on all the corrupted MNIST test set. In stage 3, to adapt to a new digit, we annotate a limited number of images as exemplary set for adaptation and use all the (corrupted) MNIST training set from that digit. We then test the agent on all (corrupted) MNIST test set of the same digit. There is no overlap between the training and test set used in stage 3.

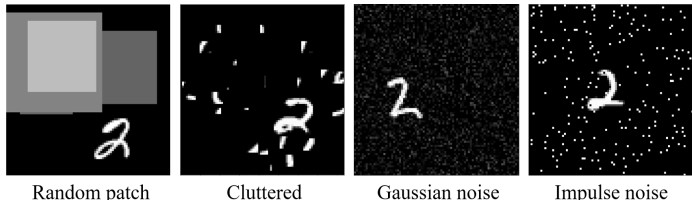

| Random patch | Cluttered | Gaussian noise | Impulse noise |

Figure 7: Corrupted MNIST Datasets: $28 \times 28$ digits on four kinds of $84 \times 84$ noisy background.

**CUB Dataset.**  In stage 1 and 2, we train on 15 warbler classes with class id between 158 and 172. There are 896 images in total. Then test the models on 5 new warbler classes with class id between 178 and 182, resulting in 294 images in total. In stage 3, the number and class ids of images for each class are presented in Table 11. We also randomly select limited number of images as exemplary set and use all unlabled data for adaptation. The $CorLoc$ is calculated using all the images of this class.

Table 11: Number of images for training and testing in stage 3 on CUB dataset.

|  | warbler | wren | sparrow | oriole | kingfisher | vireo | gull |
|---|---|---|---|---|---|---|---|
| cls id | [178, 182] | [193, 197] | [114, 119] | [95, 98] | [79, 83] | [151, 157] | [59, 64] |
| #images | 294 | 299 | 357 | 239 | 300 | 409 | 359 |

**COCO Dataset.**  For the results of Figure 3, we train on one of the five classes: *cat, cow, dog, horse, zebra*, then adapt to another four classes. The detailed number of each class for training and testing in stage 1 and 2 is shown in Table 12.

Table 12: Number of images for training and testing in stage 1, 2 on COCO dataset.

|  | cat | cow | dog | horse | zebra | elephant | giraffe | laptop | sheep | suitcase |
|---|---|---|---|---|---|---|---|---|---|---|
| train | 3619 | 649 | 3701 | 1524 | 611 | 973 | 1146 | 2844 | 260 | 1402 |
| test | 167 | 27 | 148 | 68 | 21 | 32 | 40 | 163 | 18 | 61 |

In stage 3, the agent is tested on new classes in target domain, within which we annotate limited number of images for adaptation. In comparison with few-shot object detection experiment, the models in stage 1 and 2 are trained using one single class of the 5 classes: elephant, sheep, giraffe, laptop and suitcase. Then being adapted to the four classes in Table 4. Note that the five classes are in the base classes, and the four classes are in the novel classes used in other methods. Thus, it's

harder to transfer from one single class to another due to scarcity of training data and training class. Table 4 reports the average $CorLoc$ from the five classes to each target class. We also provide the results of each source class in Table 13. From this table, we can see that transferring from related classes with target class usually performs better. For example, the $CorLoc$ from laptop and suitcase are lower than other three animal classes, especially before adaptation. After adaptation, the gap becomes smaller.

Table 13: $CorLoc(\%)$ Adaptation comparison with other methods on COCO dataset per class results.

|  | before adapt | | | | after adapt | | | |
| --- | --- | --- | --- | --- | --- | --- | --- | --- |
|  | cat | dog | cow | horse | cat | dog | cow | horse |
| elephant | 41.25 | 38.45 | 59.94 | 54.27 | 48.00 | 45.66 | 65.64 | 59.19 |
| giraffe | 17.91 | 19.18 | 22.19 | 26.25 | 42.89 | 31.88 | 35.59 | 53.68 |
| laptop | 13.87 | 4.57 | 3.24 | 1.58 | 34.93 | 15.67 | 29.43 | 20.34 |
| sheep | 29.10 | 34.13 | 36.98 | 33.14 | 46.01 | 36.64 | 49.92 | 46.26 |
| suitcase | 12.99 | 6.43 | 11.71 | 4.99 | 37.99 | 37.56 | 43.14 | 25.98 |

**Selective localization.** We investigate the agent's ability in localizing the object specified by the query set, when the set of images have two common objects. We use random patched MNIST, where each image has digit 3 and digit 4. First, the RoI encoder and projection head are trained with an additional contrastive loss to enlarge the distance between the two digits in embedding space,

$$loss_{embed} = loss_{rec} + \lambda_{trip} \cdot loss_{trip} + \lambda_{contr} \cdot loss_{contr}, \tag{7}$$

where $loss_{trip} = loss_{trip_3} + loss_{trip_4}$, learning two local ordinal structure around each class center in embedding space. We set the margin for both triplet losses as 10, and the margin for contrastive loss as 320 heuristically. We found that the larger the gap between the two margins, the better the performance. Detailed results can be found in Table 17 and its related discussion. After learning the RoI encoder and projection head, we train the agent with the reward defined with Eq. 4 in Sect. 3.3, where **c** is the prototype embedding of the targeted digit exemplary set (exemplary size is 5).

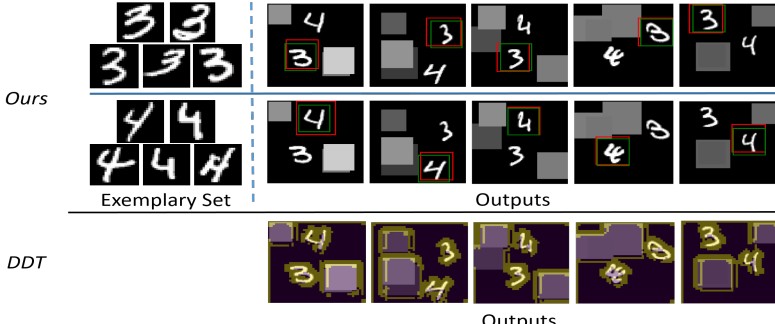

Figure 8: Selective localization vs. co-localization on two-digits data with random patch background.

Figure 8 shows an illustrative example of query object localization with task adaptation. Given two common objects coexisting in the image environment, the agent is trained to pick up one from the two during training, while quickly adapt to pick another one during testing. Different from unsupervised co-localization approaches (Wei et al., 2017) relying on low-level image clues to distinguish foreground and background, our learning objectives can be specified explicitly by switching the train/test exemplary sets. With test-time adaptation, our agent can handle even partially conflicting learning objectives.

# B    ADDITIONAL EXPERIMENTS

## B.1    ABLATION OF ORDINAL EMBEDDING AND RL

**Setting and main results on corrupted MNIST dataset.**    We analyze the effectiveness of using ordinal embedding in terms of representation and reward first on synthetic datasets of corrupted MNIST digits under cluttered, random patch, Gaussian, or impulse noises (see Figure 7 in Appendix for examples). We want to find the answers to two main questions listed below:

1. Is the embedding distance based reward signal as effective as the IoU counterpart?
2. Does the ordinal triplet loss benefit the state representation learning?

Accordingly, we consider baselines in which embeddings are trained with only autoencoder or jointly with the ordinal projection head. Besides, we also compare our embedding distance based reward against the IoU based reward used in Caicedo & Lazebnik (2015). Without loss of generality, the agent is trained on images of digit 4, and tested on images from all rest classes. In particular, we are interested in evaluating the sample efficiency of different approaches. The results under different training set sizes are shown in Figure 9. To distract the localization agent, cluttered noises with random $6 \times 6$ crops of digits are added to the background. All comparisons are based on the policy gradient (PG) method. With ordinal embedding present in both the representation and reward ("AE+Ord+Embed"), our model performance is consistently better than other settings, especially when the training set size is small. The benefit of a triplet loss is demonstrated by the comparison between "AE+Ord+IoU" and "AE+IoU". Intuitively, the similarity to a queried object is also a more natural goal than objectives defined by bounding boxes IoU, which is used in both generic object detection or localization and previous RL-based approaches.

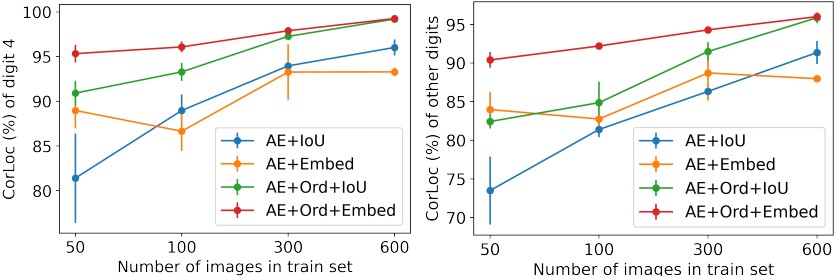

Figure 9: Comparison under different training set sizes. "AE+IoU": autoencoder, IoU based reward; "AE+Embed": autoencoder, embedding distance based reward; "AE+Ord+IoU": autoencoder and ordinal projection head, IoU based reward; "AE+Ord+Embed": autoencoder and ordinal projection head, embedding distance based reward. (a) Test performance on the trained digits $4$, (b) Average test performance on the other nine digits.

**Signed reward or not.**    Another intriguing question is that other than benefiting state representation learning, whether the embedding-based reward itself can bring any additional benefits than the IoU-based reward. From Figure 9, there is a large gap between "AE+Ord+Embed" and "AE+Ord+IoU", especially when the training set size is small. This is counter-intuitive as the ordinal reward is trained with supervision from IoU in embedding space, supposably it shall not bring any additional information. To further analyze this phenomenon, we take the sign operation off in Eq. 1 and retrain the agent. In an experiment with $50$ training images, without sign operation, the localization accuracy increases from 90.92% to 94.36% on test digit $4$, and from 82.43% to 88.64% on other digit test set. Perhaps, this improvement can be attributed to more informative feedback provided by the continuous-valued, unsigned reward.

**On policy vs. off policy.**    Many deep RL approaches are in favor of using deep Q-Network (DQN) to train an agent. Different from Caicedo & Lazebnik (2015) and Jie et al. (2016), we apply Policy Gradient (PG) to optimize it. Besides, we adopt a top-down search strategy through a RNN, while they used a vector of history actions to encode memory. We evaluate these design choices with four baselines, with "AE+IoU" setting, and trained on the same 600 sampled cluttered digit 4 images. As

Table 14 shows, the agent achieves the best performance with "PG+RNN". We find that empirically, with history action vectors the accuracy becomes worse when the agent is trained by DQN.

Table 14: $CorLoc(\%)$: DQN (with/without history action vector) vs. PG (with/without RNN)

| Method | DQN | DQN+History | PG | PG+RNN |
|---|---|---|---|---|
| Digit 4 | $88.80_{1.6}$ | $86.54_{4.3}$ | $88.98_{2.9}$ | $\mathbf{94.68_{0.9}}$ |
| Other digits | $84.21_{2.0}$ | $81.75_{3.4}$ | $81.91_{2.7}$ | $\mathbf{89.05_{1.7}}$ |

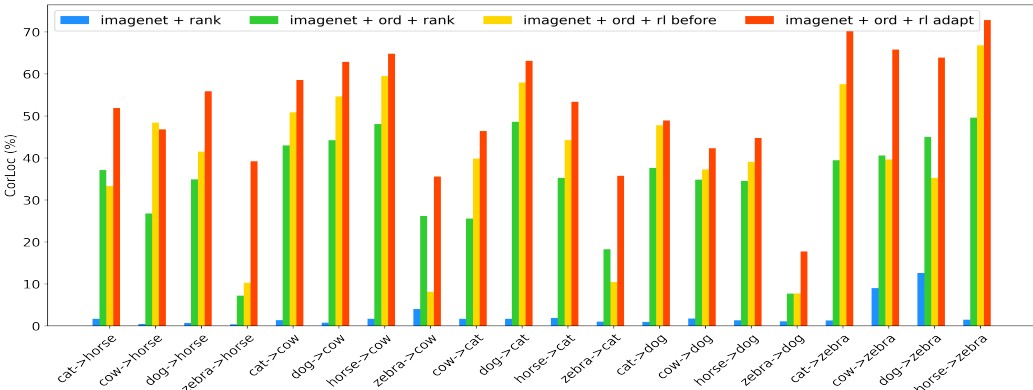

Figure 10: $CorLoc(\%)$ comparison with ranking method using ImageNet pre-trained backbone.

**More results on COCO dataset.** Figure 10 provides results using ImageNet pre-trained VGG-16 network as backbone with the same training strategy as Figure 4. To further demonstrate the effectiveness of ordinal embedding, we compute the Spearman's Rho rank correlation between embedding distance to the prototype and IoU to the ground-truth. The results are shown in Figure 11. Here we also add CLIP (Radford et al., 2021) pre-trained ViT as backbone for comparison. The rank correlation is smaller than $-0.7$ on all backbones with ordinal embedding, exhibiting ordinal embedding preserves the IoU order, thus is better for the ranking purpose. Although pretty effective, embedding distance is still not a perfect indicator of the ranking of IoU. Thus, directly formulate the object localization problem as a search problem leads to suboptimal localization accuracy.

Table 15: $CorLoc(\%)$ compare backbones on source domain.

| backbone | cow | cat | dog | horse | zebra |
|---|---|---|---|---|---|
| ImageNet pre-trained VGG-16 | 70.37 | 68.46 | 61.26 | 61.28 | 79.36 |
| Faster RCNN pre-trained VGG-16 | 66.67 | 72.46 | 61.59 | 60.29 | 71.25 |
| CLIP pre-trained ViT | **74.07** | **82.64** | **70.95** | **76.47** | **80.95** |

## B.2 COMPARE DIFFERENT OFF-THE-SHELF NETWORKS AS THE BACKBONE

It is interesting to study the choices of off-the-shelf pre-trained networks as the backbone, such as CLIP or supervised embedding provided by Faster RCNN or a classification network. Since these networks have been exposed to large-scale dataset, it is interesting to see whether policy adaptation is still needed. We compare different backbones on both source domain and target domain using our method. Table 15 reports the $CorLoc$ of training and testing on source domain. The large-scale pre-traind ViT backbone consistently performs the best, comparing to the other two VGG-16 models. Table 16 compares the backbones on target domain with new classes. The test-time adaptation still brings a large margin of improvement. Interestingly, we also found that the Faster-RCNN embedding offers the best performance on the target domain before adaptation, while the ViT network trained on CLIP dataset provides the best performance after adaptation, indicating different generalization mechanisms. They both outperform the ImageNet backbone initially considered.

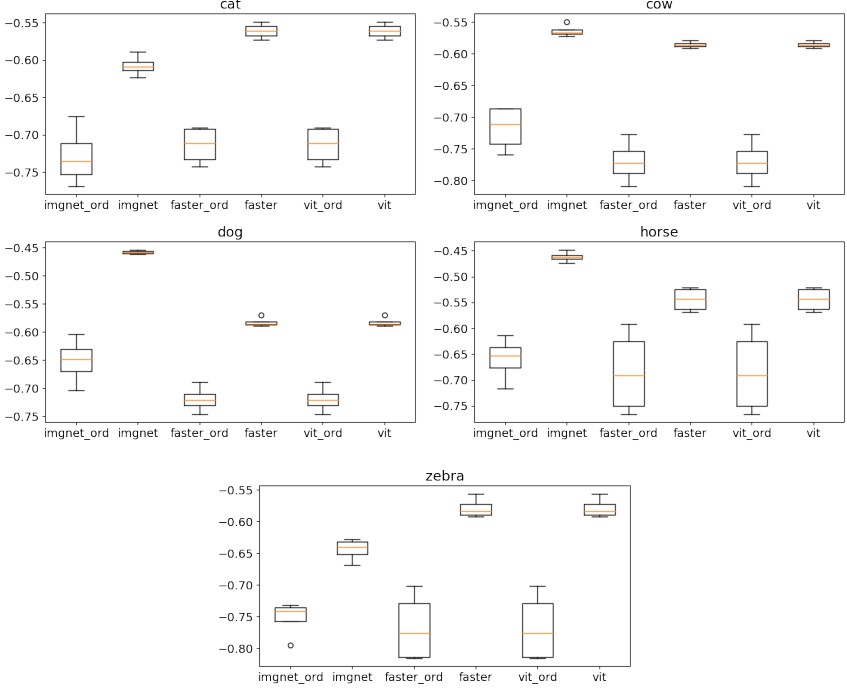

Figure 11: Rank correlations between embedding distance and IoU: using different embedding functions such as ImageNet and Faster RCNN, with or without the ordinal pre-training, and ViT.

Table 16: $CorLoc(\%)$ compare backbones on target domain.

| | before adapt | | | after adapt | | |
|---|---|---|---|---|---|---|
| | ImageNet VGG-16 | Faster RCNN VGG-16 | CLIP ViT | ImageNet VGG-16 | Faster RCNN VGG-16 | CLIP ViT |
| cat->horse | 33.32 | **35.50** | 18.42 | 51.89 | 47.64 | **56.41** |
| cow ->horse | 48.41 | **54.55** | 53.67 | 46.80 | 59.61 | **63.06** |
| dog ->horse | 41.50 | **46.48** | 15.70 | 55.89 | 56.83 | **58.62** |
| zebra ->horse | 10.29 | **16.86** | 6.74 | 39.22 | 34.19 | **46.39** |
| cat ->cow | **50.85** | 42.99 | 36.26 | **58.58** | 53.26 | 55.52 |
| dog ->cow | 54.63 | **58.65** | 43.50 | 62.86 | **64.15** | 58.76 |
| Horse ->cow | 59.52 | **61.32** | 52.54 | 64.83 | 65.23 | **68.16** |
| Zebra ->cow | 8.14 | **11.92** | 7.19 | 35.56 | 38.26 | **52.65** |
| cow ->cat | 39.84 | **47.39** | 38.79 | 46.42 | 51.15 | **61.67** |
| dog ->cat | 57.97 | 63.84 | **66.60** | 63.12 | 65.18 | **76.83** |
| horse ->cat | 44.25 | **47.67** | 27.80 | 53.39 | 52.96 | **63.87** |
| zebra ->cat | 10.45 | **17.67** | 2.47 | 35.73 | 31.40 | **49.12** |
| cat->dog | **47.81** | 45.61 | 49.69 | 48.94 | 49.83 | **61.75** |
| cow ->dog | 37.28 | **37.64** | 30.13 | 42.33 | 37.10 | **50.94** |
| horse ->dog | 39.07 | **40.76** | 23.89 | 44.77 | 40.69 | **55.68** |
| zebra ->dog | 7.74 | **11.83** | 2.88 | 17.73 | 30.64 | **36.48** |
| cat ->zebra | **57.58** | 15.82 | 22.59 | **70.28** | 45.83 | 69.39 |
| cow ->zebra | 39.64 | **60.55** | 37.75 | 65.80 | 64.21 | **72.18** |
| dog ->zebra | **35.27** | 18.25 | 15.33 | 63.91 | 58.16 | **67.59** |
| horse ->zebra | **66.82** | 56.63 | 61.37 | 72.83 | 68.74 | **75.01** |

## B.3 THE EFFECTS OF MARGIN

The margin in triplet loss is selected heuristically. It is not sensitive except in the selective localization experiment (Figure 8), where there are two different digits in each image. For this experiment, we trained two ordinal structures around each digit using triplet loss with margin $m_1$, and add additional contrastive loss with margin $m_2$ to separate the centers of the two different digits as far as possible. And we found out that the model works best when $m_2 \gg m_1$. In our experiment, we set $m_1 = 10, m_2 = 320$. The results of using different set of $m_1$ and $m_2$ are presented in Table 17.

Table 17: Results of different margin configuration in selective localization.

| $m_1$ | 10 | 10 | 10 | 10 | 10 |
|---|---|---|---|---|---|
| $m_2$ | 60 | 70 | 80 | 160 | 320 |
| $CorLoc(\%)$ | 86.54 | 87.92 | 88.32 | 91.39 | 98.52 |

## B.4 SIZE OF TRAINING SET

In this experiment, we train on class giraffe in stage 1 and 2, then adapt to cat, cow, dog and horse. We set the training set size as [200, 500, 700, 1146], exemplary set size as 5. We compare our after adaptation results with TFA w/ fc (Wang et al., 2020) *one-way 5-shot* setting, where their model is trained on all 60 base classes, while ours is only trained on one of the base classes. Figure 12 shows the results, in which the dotted lines are the results of TFA w/ fc. Ours performs much better than their method. Except on cat class, ours is better than theirs with only 200 images for training.

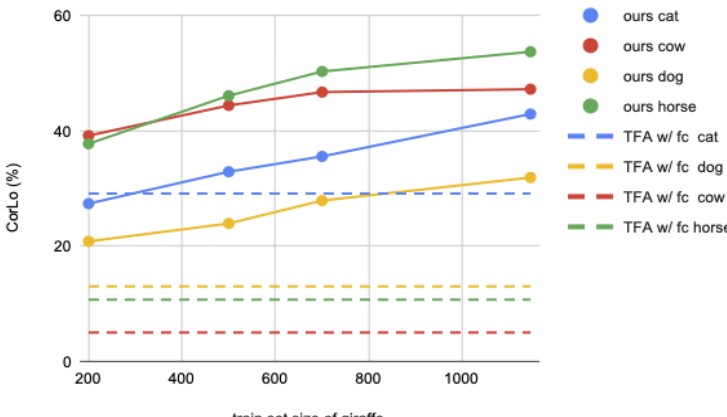

Figure 12: Results using different training set size in stage 1, 2.

## B.5 SIZE OF EXEMPLARY SET

We also compare the effect of different size of exemplary set during training and adaptation on CUB-warbler dataset. During training stage, we use shuffle proto training strategy, and set exemplary set size as 2, 5, 15, 25. The results without adaptation on test set are in Table 18. Both $OrdAcc$ and $CorLoc$ increase with exemplary set size. For adaptation stage, the range of exemplary set size is from 2 to 200. And the results are in Table 19. The test performance does not increase much with the exemplary set size. One possible explanation is that the data points in embedding space are compact, thus prototype doesn't change much when increasing exemplary set size. We will analyze the influence of multiple prototypes per class in future experiments.

## B.6 PROTOTYPE SELECTION

We further evaluate the choice of anchor in the triplet loss for both the pre-training of state representation and the ordinal reward for the training of agent. We study $(i)$ whether ordinal embedding can be

Table 18: Effect of exemplary set size during training stage.

| Size | $OrdAcc(\%)$ | $CorLoc(\%)$ |
|------|--------------|--------------|
| 2 | $94.39_{\pm1.7}$ | $84.18_{\pm6.5}$ |
| 5 | $94.83_{\pm2.0}$ | $88.10_{\pm0.2}$ |
| 15 | $95.69_{\pm1.7}$ | $89.22_{\pm1.9}$ |
| 25 | $93.82_{\pm1.0}$ | $89.64_{\pm2.3}$ |

Table 19: Effect of exemplary set size during adaptation.

| Size | 2 | 5 | 50 | 100 | 150 | 200 |
|------|---|---|-----|-----|-----|-----|
| $CorLoc(\%)$ | $89.12_{\pm1.9}$ | $89.67_{\pm1.1}$ | $90.15_{\pm0.8}$ | $90.36_{\pm0.5}$ | $89.63_{\pm0.2}$ | $90.14_{\pm0.5}$ |

trained in reference to an anchor from a different image instances, $(ii)$ whether it is advantageous to use the prototype embedding of an exemplary set, rather than instance embeddings, and $(iii)$ whether mimicking the test condition in training yields any improvement.

Table 20: Anchor choice comparison. Note that in evaluations the $OrdAcc$ is always computed using instance as anchor.

| Mode | $OrdAcc(\%)$ | $CorLoc(\%)$ |
|------|--------------|--------------|
| Self | $97.2_{\pm0.7}$ | $61.0_{\pm2.0}$ |
| Proto | $95.2_{\pm1.6}$ | $77.9_{\pm0.4}$ |
| Shuffle self | $92.4_{\pm1.4}$ | $73.8_{\pm2.5}$ |
| Shuffle proto | $96.2_{\pm1.5}$ | $88.1_{\pm0.2}$ |

We use the CUB-Warbler dataset with more foreground background variations than the corrupted MNIST dataset. The training and test set contains 15 and 5 disjoint fine-grained classes respectively, resulting 896 images for training (viewed as a single class) and 294 for testing. Table 20 shows the $OrdAcc$ and $CorLoc$ in four settings. "Self" uses the embedding from images cropped by the ground-truth box from the same instance; "Shuffle self" uses the ground-truth box cropped image emebedding from a different instance; Similarly, "Proto" uses the prototype of a subgroup containing the training instance within the same batch; "Shuffle proto (SP)" uses the prototype of a subgroup from a different batch without the training instance. Results suggest that this training strategy brings compactness to the training set, constructing an ordinal structure around the cluster. For "Shuffle proto", while the $OrdAcc$ is lower than others, its $CorLoc$ is the best with large margin. Matching the condition between training and testing indeed improves generalization to new classes on this dataset.

## B.7 ANNOTATION REFINEMENT

Annotation collection plays an important role in building machine learning systems. It is one task that could benefit greatly from automation, especially in cost-sensitive applications. The human labeling efforts can be reduced, not only in terms of the number of annotate samples per class, number of annotate classes (one-class at the minimum), but also the level of accuracy required. We explore the direction of iterative refinement of annotation quality with minimal human guidance. Given an agent trained extensively with inexact human annotations, can we adapt it to refine the annotation with a few well-annotated examples? We purposefully enlarge the ground truth bounding boxes in the corrupted MNIST and CUB datasets, and adapt the trained agent using the original ground truth box in an exemplary set of size 5.

For corrupted MNIST dataset, the agent, RoI encoder and projection head are first trained with 50 digit 3 images with loose ($38\times38$ ) bounding box annotation. Then the agent is adapted to digit 1 images, where the exemplary size=5, and the ground-truth boxes are $28\times28$. For CUB dataset, during training, the ground-truth box in each resized $224\times224$ image is expanded 10 pixels each side. During adaptation we use the original ground-truth boxes. In adaptation from entire body to head experiment on CUB dataset, we train on original entire body ground-truth bounding box. During adaptation, 15 randomly sampled exemplar images are annotated with head bounding box. The agent

is trained to adapt to localize head by minimizing the distance between embedding of predicted box and prototype of head embeddings. The results are shown in Figure 13 (left panel). Preliminary results with agent trained on the entire bird body and adapt to bird head are shown in Figure 13 (right panel), with a few successful cases on the first row and failure cases on the second row.

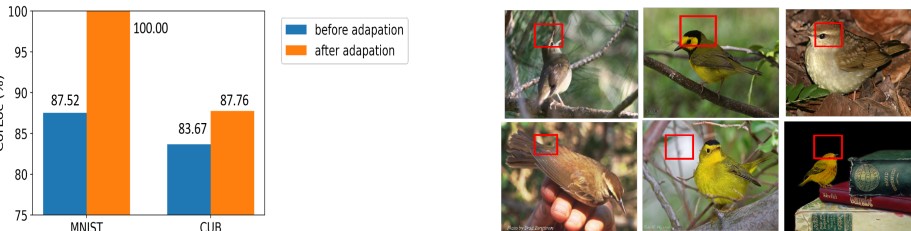

Figure 13: Annotation refinement. **Left:** Loose to tight transfer. **Right:** entire body to head transfer.

### B.8    FEW-SHOT LOCALIZATION ON OMNIGLOT

The proposed ordinal reward signal also makes our approach amenable to few-shot training, when only a small subset of training images per class are annotated. Different from the transfer learning setting, in few-shot setting limited annotations across multiple classes are available during training. The ordinal reward can be viewed as meta information. We evaluate our method under few-shot setting on corrupted Omniglot dataset (Lake et al., 2015) and CUB-warbler dataset. For Omniglot, We put each $28 \times 28$ character in $84 \times 84$ random patch background. The train and test set contains 25 different classes respectively, thus 500 images for each set. We randomly sample 100 iterations for training and testing. For CUB-warbler datset, as we did in Sect. 4.2 we train on the same 15 species from the "Warbler" class, and adapted to 5 new species of "Warbler", thus 896 and 294 images respectively. We randomly sample 100 and 50 iterations for training and testing. We use 5-shot 5-way, set exemplary set size as 5, and use proto training strategy for both dataset. The results are shown in Table 21. As an implicit meta learning method, our approach achieves $99.94\%$ and $90.52\%$ $CorLoc$ on the two datasets. We can also leverage explicit meta learning method, such as MAML (Finn et al., 2017) to further improve the results. We will leave this part as future work. Although initial results are promising, more efforts are needed to validate whether the proposed RL approach can achieve state-of-the-art performance, but it is beyond the scope of this work.

Table 21: Evaluation under few-shot localization setting.

| Dataset | $OrdAcc(\%)$ | $CorLoc(\%)$ |
|---|---|---|
| Omniglot | $95.11_{\pm 0.6}$ | $99.94_{\pm 0.1}$ |
| CUB-warbler | $91.28_{\pm 0.9}$ | $90.52_{\pm 0.7}$ |

## C INTERMEDIATE RESULTS

The agent tends to predict expected actions to transform the box so that it can finally get to the target location. It can also be observed from Figure 14 that the embedding distance between predicted boxes and ground-truth boxes is in agreement with the corresponding IoU. This further demonstrates the effectiveness of the localization agent.

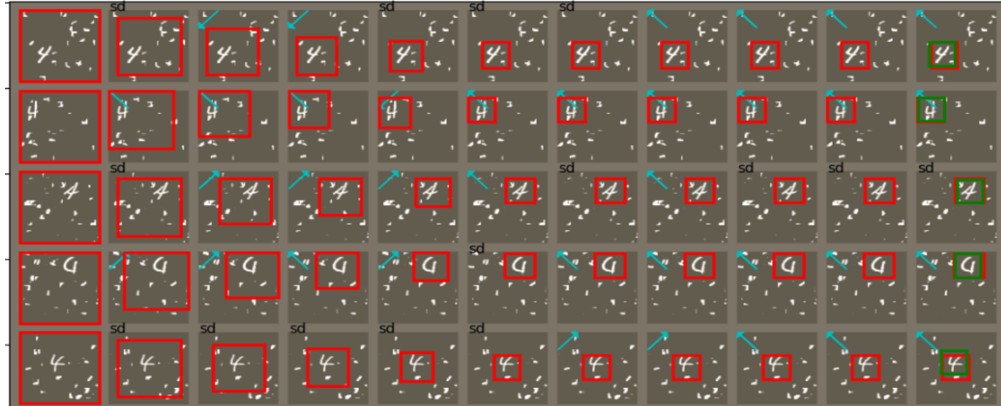

Figure 14: Starting from the whole image, the agent is able to localize digit 4 within 10 steps. Each row shows localizing one image, and each column is a time step. Green box means ground-truth bounding box.

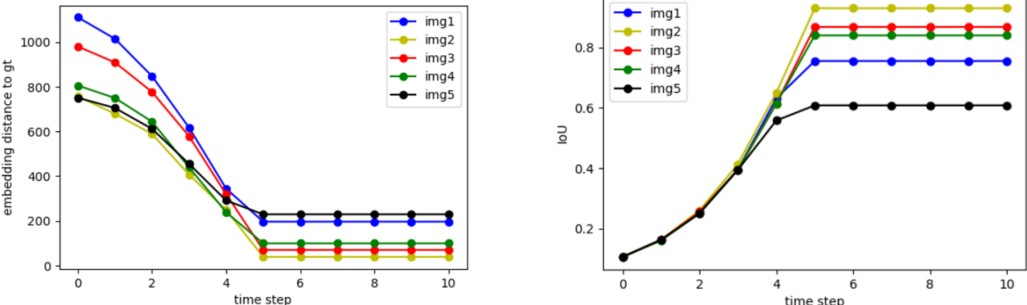

Figure 15: The IoU relation between ground-truth and predicted box is accurately represented in embedding space. **Left:** the embedding distance between ground-truth and predicted box in each step. **Right:** the IoU between ground-truth and predicted box in each step.

