# OpenReview forum: "Learning Transferable Reward for Query Object Localization with Policy Adaptation"
_ICLR.cc/2022/Conference — ICLR 2022 Poster_

### Official Review · Reviewer_DiRh · 2021-11-01

**Correctness:** 4
**Technical Novelty And Significance:** 2
**Empirical Novelty And Significance:** 3
**Recommendation:** 8
**Confidence:** 2

**Main Review:**


This paper formulates the query object localization problem as an RL task. The task itself has many applications in other fields such as robotic manipulation. Prior work directly uses IoU as the reward measure for training localization agents while this paper proposes to first learn an embedding space that respects the IoU-based ordering and then directly use the distance between embedded features as the reward signal for training the localization agent, i.e. the more similar a state is with the query image in the embedding space, the higher the reward. Experiments on several query object localization tasks in MNIST, CUB and COCO datasets demonstrate the proposed method’s ability to transfer learned rewards from one task to another.

One question I have is on the evaluation protocol. The authors report the percentage of correct localization as the evaluation protocol (which counts a localization as correct if IoU is greater than 0.5). Why not use standardized measures such as mean IoU or mean average precision? How does CorLoc compare with mAP iou=0.5? What is the advantage of using CorLoc?

The authors mention that the method can be easily modified to deal with multiple queries objects or no queried object in the image, it does not seem to be trivial for readers that are not familiar with this task setting and it would be great if the authors could elaborate on this.

The baseline methods compared in Table 6 are not clearly explained and it can be hard to see whether the comparison is fair/meaningful. It would help a lot for understanding the results if the authors could discuss the differences between the baseline methods and the proposed algorithm under test.

One interesting baseline/ablation I think could be added here is to use an off-the-shelf pre-trained network as the embedding model/RoI encoder. Large-scale models such as CLIP could be tested here to see if these large-scale models offer better representation for transfer than task-specific ones learned in the proposed way.



**Summary Of The Paper:**

This paper presents a new metric learning method on the RL formulation of query object localization. The metric learning method is based on a contrastive learning formulation of ordinal embeddings, which are pretrained with data augmentations and a loss formulation that respects IoU orderings. The experiment results have demonstrated that the proposed embedding metric increased the task performance compared with the baseline metric (IoU), and shown better generalization behavior on novel image categories.


**Summary Of The Review:**

This paper presents a novel method for solving query object localization as an RL task. The authors propose to learn an embedding space that respects the IoU ordering of bounding boxes and then use the distance in the embedding space as reward signal instead of raw IoU scores. The writing and presentation of the paper is clear. However, it is not easy to understand the similarities and differences between the proposed method and the baselines it compared with in the transferring task. It would better help the audience understand the significance of the results if the baseline methods are explained with more details.

---

> ### Author Response · Authors · 2021-11-22
> **Response**
>
> We thank Reviewer DiRh for the valuable feedback and we appreciate the positive comments.
>
> Q1. One question I have is on the evaluation protocol. ...Why not use standardized measures such as mean IoU or mean average precision? How does CorLoc compare with mAP iou=0.5? What is the advantage of using CorLoc?
>
> Thank you for this question. Please refer to "common response" - evaluation protocol.  Mean IoU is more used as an evaluation metric for semantic image segmentation. mAP is more used in object detection. CorLoc is calculated with the final predicted bounding box. While mAP is the mean average precision across multiple classes where we only have one class. And AP is calculated is more suitable for methods outputting multiple bounding boxes, yet we only have one output as the final prediction.
>
> Q2. ...modified to deal with multiple queries objects or no queried object in the image, it does not seem to be trivial...
>
> We've added some discussions on one possible way of extension (Section 5). Please refer to "common response" - extensions to multiple objects.
>
> Q3. The baseline methods compared in Table 6 are not clearly explained ... whether the comparison is fair/meaningful. ... ifferences between the baseline methods and the proposed algorithm under test.
>
> Sorry for the confusion. Please refer to the "common response" - Transfer learning vs. Few-shot setting. The purpose is to compare two different configurations of training tasks (one-way vs. many-way few-shot). The test setting is the same "one-way 5-shot setting" for all the approaches. We used pretrained models provided by the official code. They trained on 60 base classes. While ours is trained on five base classes: elephant, sheep, giraffe, laptop and suitcase. They finetuned on all 80 novel classes (including the 4 classes in Table 6. under one-way 5-shot setting). While we also use only five images as exemplary set for adaptation. The test setting is comparable with few-shot methods. We've added more details about the experiment design in Section 4.3 Table 6 (Table 5 in updated version) and appendix A.2, COCO dataset.
>
> Q4. One interesting baseline/ablation I think could be added here is to use an off-the-shelf pre-trained network ... such as CLIP could be tested here...
>
> Thanks for this great suggestion! We have implemented a ViT network trained on CLIP. The performance outperforms the ImageNet network we used initially, but the test-time adaptation still brings it a large margin of improvement. Results can be found in appendix B.3.

---

> > ### Comment · Reviewer_DiRh · 2021-12-01
> > **Thanks for the authors' response**
> >
> > I appreciate the authors' response and most of my questions have been addressed. I'm happy to see the updates to the paper and will adjust my evaluation to reflect this.

---

### Official Review · Reviewer_pHvg · 2021-11-02

**Correctness:** 3
**Technical Novelty And Significance:** 3
**Empirical Novelty And Significance:** 4
**Recommendation:** 6
**Confidence:** 4

**Main Review:**

strengths:
* reinforcement learning approaches are interesting in this domain
* results seem promising
* experiments thorough and give some depth to the methodology

weaknesses:
* the paper is overly complicated and lacks clarity on many details.

observations:
* I cannot understand what figure 1 is meant to show. Is it the computation ? I find that very unclear. I find it insufficient to give a clear overview of the method (which i assume is the goal).
* sec. 3.1 needs a rewrite. The actions are unclear, the role of the reward should be clearly stated, the environment is not an image it's a set of states and transitions etc.
* sec. 3.2 it is unclear what box perturbation means. Are the features perturbed or just the crop ?
* \rho in section 3.2 is explained in section 3.3 What is m in eq 3 ? What is an anchor in this context ?
* in algo 1 the entropy is not added so the equation is not correct

**Summary Of The Paper:**

The paper proposes a reinforcement learning approach for localization in images. The idea is to consider and environment where the states are crops of an image, the actions are changes in the current coordinates of the current crops starting at the full image. The method consists of learning both a reward function and a policy that maximizes that reward function. The policy is learnt with REINFORCE with entropy regularisation. The manner in which the reward function is formulated and learnt is quite appealing. To allow for more flexibility at test time, the reward function is made conditional on some exemplars and learnt in a few shot manner based on state features. Using a triplet loss to ensure the reward function is increasing as the overlap is increasing is quite good.

The experiments are quite extensive and I find compelling for the most part. Ablation of the elements of the method is well executed and benefits of few short learning are compelling enough in my opinion. The results on COCO are a bit limited but I would be willing to overlook that.




**Summary Of The Review:**

The paper has enough novelty and importance but the rewriting needs a lot of work to make more precise and cohesive. With a solid improvement to clarity I would support acceptance.

---

> ### Author Response · Authors · 2021-11-22
> **Response**
>
> Thank you for the comments! We have modified the paper accordingly and rewritten the methodology part (Sec. 3.). Please let us know if this has addressed your concerns on the clarity.
>
> Q1. I cannot understand what figure 1...
>
> Sorry for the confusion. We have modified Figure 1 and added more explanations. Hopefully, it looks more clear now.
>
> Q2. Sec. 3.1 needs a rewrite.
>
> We have added more supporting details in Section 3.1 to make this paper self-contained. Actually, we consider a single image as the environment, the same formulation as active localization [1]. The agent can only take the feature of the current region as state and output actions accordingly. We've added more descriptions in the revised manuscript.
>
> Q3. Sec. 3.2 it is unclear what box perturbation means. Are the features perturbed or just the crop ?
>
> Box perturbation means crop of the image, with an IOU between 0 to 1 to the ground truth bounding box. We've clarified this in Section 3.2 box perturbation.
>
> Q4. \rho in section 3.2 is explained in section 3.3 What is m in eq 3 ? What is an anchor in this context?
>
> Sorry for the confusion. We've addressed them in the manuscript. $m$ in eq 3 is the margin parameter in the triplet loss for ordinal embedding training, by which the perturbed box closer to groundtruth in terms of IoU would also be closer in embedding space. The anchor in triplet loss is the prototype embedding, i.e., average of groundtruth embeddings. There are several choices for the anchor, also explained in Section 3.2 choice of anchor.
>
> Q5. in algo 1 the entropy is not added so the equation is not correct
>
> Sorry for the confusion. We omitted entropy loss for simplicity as usually it's not included in RL update rule. We've added it in Algorithm 1 and Table 8 for clarity.
>
> Reference:
>
> 1. Juan C Caicedo et al., Active object localization with deep reinforcement learning, ICCV 2015.

---

> > ### Comment · Reviewer_pHvg · 2021-12-03
> > **response to rebuttal**
> >
> > Thanks to the authors for the reply.
> >
> > With the answers to my questions and the new shape of the paper I think this is a worthwhile contribution to the conference. I am updating my rating to reflect this.

---

### Official Review · Reviewer_tkLp · 2021-11-03

**Correctness:** 4
**Technical Novelty And Significance:** 3
**Empirical Novelty And Significance:** 3
**Recommendation:** 6
**Confidence:** 4

**Main Review:**

Strengths:
- The idea of using a transferable reward is interesting and novel. But as explained in the weakness section, the design of an RL agent for object localization is not empirically justified.
- Experiments on few-shot object detection show good improvements over recent methods (e.g. TFA).
- The paper presents some good investigation on the design of the loss function for the ordinal embedding network (Figure 4). In addition, various RL algorithms have been tested.

Weaknesses:
- The paper has not studied the separate effect of the RL component and the ordinal embedding component. Since during ordinal embedding training the model has access to a pool of bounding box labels, the authors could use the same data to train a class agnostic bounding box proposal generator (like the first stage in Faster-RCNN), or it could leverage selective search bounding boxes. Then, with the ordinal embedding module, it can choose the most similar region in the ordinal embedding space as the final box prediction. Such a pipeline removes the complicated procedures of RL and sequential box refining, but preserves the ordinal embedding component, and should be studied as a baseline to corroborate the claim that a transferable “reward” is required for such a task. I ask this because the problem structure is static and the state will not change based on the action taken by the agent, and instead of framing it as RL, people can view it simply as a search problem.
- In addition to the baseline above, the ordinal embedding component can be replaced by a supervised embedding provided by Faster-RCNN or simply a pretrained classification network using the labeled portion of the data. These baselines will separate the effect of RL and ordinal embedding in a 2x2 experiment matrix.
- Another concern with the current RL agent setup is that it is only configured to output a single object, and cannot handle if the image contains multiple objects. This is a limitation compared to more widely adopted settings of object detection.
- As a result of the simplified setup, the paper uses the CorLoc metric to measure the percentage of images with IoU > 0.5, and it is not the same metric used in few-shot object detection (which uses mAP over different splits). The discrepancy in the evaluation protocol may make it difficult for follow-up works to compare to this work.
- Section 3.3 should be named “ordinal embedding network architecture” instead of “model architecture” since it only describes the network that provides the ordinal embedding.
- In the COCO few-shot detection experiment, although it is encouraging to see that the proposed model only needs to learn from one class, the setup seems a bit non-standard and not directly comparable to few-shot learning baselines (it’s making the task harder for the proposed method). It would be good to investigate the amount of training data needed, by testing on a varying number of training classes. Perhaps this can show that the proposed transferable reward is more data efficient and more generalizable.

**Summary Of The Paper:**

This paper proposes a new method for the task of query object localization. It learns an embedding network such that the image crop that is closer to the query object will have a closer embedding distance as well, and the improvement of the embedding distance will be used as the reward function. The RL agent is then employed to maximize the reward. The paper found that the proposed approach compares favorably to DDT on object location on CUB and FTA on few-shot object detection on COCO.

**Summary Of The Review:**

The paper presents an interesting idea for object localization using RL. My main concern is that the current experimental designs are weak, because of 1) less sufficient studies on the effect of individual components, and 2) non-standard object detection protocol. Therefore my initial rating is “weak reject”.

After seeing the authors' response, I decided to increase my rating from 5 to 6 since the added experiments addressed my concerns.

---

> ### Author Response · Authors · 2021-11-22
> **Response**
>
> Thank you for all the comments and constructive suggestions. We address the concerns below.
>
> Q1: The paper has not studied the separate effect of the RL component and the ordinal embedding component. ...it could leverage selective search ...should be studied as a baseline...people can view it simply as a search problem.
>
> Thank you for the great suggestions on this ablation study and the possible baselines. Replacing the RL component with search helps justify the necessity of the RL-based localizer. We have implemented them and the results can be found in appendix B.1.
>
> This problem can be formulated as a search problem. We considered a number of off-the-shelf backbone networks as the embedding. We found that pre-training with the proposed ordinal loss significantly improves the rank consistency of these backbone networks.
>
> Although pretty effective, embedding distance is still not a perfect indicator of the ranking of IoU. We found the localization accuracy of ranking by embedding distance to the prototype is suboptimal. While the RL localization agent shows more tolerance to the defects of the reward function. In fact, the RL approach can be viewed as an efficient solution to the search problem in an unlimited box proposals space, requiring only a limited number of evaluations.
>
> Q2. ... the ordinal embedding component can be replaced by a supervised embedding ...or simply a pretrained classification network ...separate the effect of RL and ordinal embedding in a 2x2 experiment matrix.
>
> We considered the suggested pre-trained embeddings as backbone networks and performed additional ablation studies, with or without the ordinal component. Results are reported in appendix B.1 and B.3. Localization accuracies on both source and target domains are evaluated. We found the ordinal component critical. Interestingly, we also found that the Faster-RCNN embedding offers the best performance on the target domain before adaptation, while the ViT network trained by CLIP provides the best performance after adaptation, indicating different generalization mechanisms. They both outperform the ImageNet backbone we considered initially.
>
> In addition, we also considered a head-to-head comparison against Faster-RCNN (appendix B.2), both trained on one class and adapt to a different class. Results on the source domain (Table 13) show that Faster-RCNN yields very strong performance.  But on the target domain, Table 14 shows the advantage of the proposed test-time policy adaption over the traditional fine-tuning of Faster-RCNN. Note that fine-tuning requires image-box pairs in the target domain while our policy adaptation approach does not.
>
> Q3. Another concern with the current RL...only output a single object, and cannot handle if the image contains multiple objects...
>
> We admit that it is a limitation. Please refer to the "common response" part -  Extension to detect multiple objects
>
> Q4. As a result of the simplified setup, the paper uses the CorLoc ...not the same metric used in few-shot object detection...
>
> Although our approach is highly related to object detection, the setting is not exactly the same.  Please refer to  the "common response" part  -  Evaluation protocol
>
> Q5. Section 3.3 should be named “ordinal embedding network architecture” instead of “model architecture” since it only describes the network that provides the ordinal embedding.
>
> Thanks for pointing it out. We've corrected it in the manuscript.
>
> Q6. In the COCO few-shot detection experiment..the setup...non-standard and not directly comparable ...to investigate the amount of training data needed...
>
> Please refer to the "common response" - Transfer learning vs. Few-shot setting.  We provide more details in appendix A.2 COCO dataset, and more detailed results in Table 11. The few-shot detection experiment is meant to compare the different choices of data collection in training:  one-way vs. many-way few-shot, and evaluated under the same "one-way five-shot" test setting. We used pretrained models provided by the official code. They trained on 60 base classes. While ours is trained on five base classes: elephant, sheep, giraffe, laptop and suitcase. They finetuned on all 80 novel classes (including the 4 classes in Table 6. under one-way 5-shot setting). While we also use only five images as exemplary set for adaptation. The test setting is comparable with few-shot methods. We compared train set size as suggested in appendix B.5, from where it can be seen our model is data-efficient.

---

> > ### Comment · Reviewer_tkLp · 2021-12-06
> > **Re: Response**
> >
> > I thank the authors for their response and the additional experiments studying the effect of RL and different pretrained embeddings. They address my concern. However, the added experiments seem to only study the setting of one training class transferring to one test class and potentially they could be enhanced by considering the benefit of training on more classes.

---

### Official Review · Reviewer_yBs4 · 2021-11-03

**Correctness:** 3
**Technical Novelty And Significance:** 3
**Empirical Novelty And Significance:** 3
**Recommendation:** 8
**Confidence:** 3

**Main Review:**

Overall, I found this work interesting (especially as I wasn’t aware of RL-based object localization), and it covers a rather specific problem setting well.

It is quite hard to find details of decisions and specific model choices however, so perhaps the authors can improve on that and clarify some of these points:
1. The early sections provide a good overview of the problem setting and of the particular way this will be addressed. However, some details of exactly what modules are trained, with which losses, under which data distribution are lacking. I also found myself confused when understanding exactly how the policy was set up and what it was receiving.
   1. Figure 2 is clear, but Algorithm 1 does not reuse the same modules clearly. Figure 9 in the Appendix was also quite helpful, but the ROI encoder outputs 3 vectors in Figure 2, hence it isn’t directly clear what is happening. I understand you probably just use the Encoder with a RoIAlign using the changing bounding box, but this should be clarified.
   2. What is trained and frozen in the different tasks? For example in Figure 4, was the Encoder/L_triplet trained on just digits 4 or everything?
   3. Network and loss details are hidden in the text in random places, could you add a clear table in the Appendix for all of these?
   4. There is no mention of how the margin `m` was chosen.
2. As a more meta-discussion point: why is it beneficial to treat this as a RL problem, instead of “just” outputting the whole sequence of actions in one go?
   1. i.e. consider using a sequential action model which just spits out a variable number of bounding box modification actions, and treat the problem as a bandit setup. Could you discuss and contrast why that isn’t appropriate?
3. The way that perturbed bounding boxes (one good, one bad) are sampled for a given image wasn’t extremely easy to find (it is mentioned without details in 3.2, then again in 4.1, with perhaps alternatives considered per dataset if I understood the Appendix well).
   1. It might be useful to spell this out clearly in the Appendix.
   2. I would expect this to have a rather strong effect, did you find it to be sensitive?
4. Figure 4 uses 50 images in the training set as the smallest number.
   1. Was the encoder trained on 50 images as well?
   2. Did you try using fewer? It would make a stronger point if your method could still cope well.
   3. How dependent is this on the size of the examplar set?
5. Did you explore other ways to use the exemplar set, instead of just averaging them into a prototype?
   1. For example, Transformers or other set-invariant modules?
6. The “signed reward or not” section is not the clearest. There is a simple explanation for why “removing the sign operation” is helpful, which is that it gives continuous (instead of binary -1/+1) rewards at every steps, which makes the overall RL problem easier as the rewards provided are denser and more informative?
7. What is the loss_embed presented in Appendix A.2.1 and is it used for any results apart from the selective localisation one? It is more complicated and less general than the loss present in the main text (e.g. it is aware that only 2 digits will ever be present in an image).


**Summary Of The Paper:**

This paper extends works performing RL-based object localization by:
1. Conditioning the localization on a exemplary set of images, instead of more classical hardcoded finite set of classes
2. Introducing a new reward signal which doesn’t explicitly use IoU but instead enforces a rank-preserving metric space (where distances reflect how well 2 bounding boxes match up).

This is shown on multi-MNIST data, CUB and COCO. They are targeting rather challenging generalisation setups, where a policy is only trained to localize 1 digit class (in the MNIST case say) and extrapolates to all other ones.


**Summary Of The Review:**

Overall, I find this work relevant and well executed, although some details of how novel it is and details of its implementation might deserve to be improved. Results are good on toy domains and acceptable on COCO, but it is unclear how well it would cope with more complex situations. For now, I’d qualify this as borderline and tend to accept given it was an interesting read, but I am not extremely familiar with the literature and related work.

---

> ### Author Response · Authors · 2021-11-22
> **Response--Part 1**
>
> Thank you for the valuable comments! We've added more discussion on the decisions and specific model choices in the manuscript and provided more details in the appendix as requested. We will address your specific questions below.
>
> Q1. ...the ROI encoder outputs 3 vectors in Figure 2, hence it isn’t directly clear what is happening...
>
> Sorry for the confusion. We've updated Figure 2 and provided more details in appendix A.1. Your understanding is correct.  The output of RoI encoder is the feature of each RoI,  and they are combined in one tensor actually. In Figure 2, we plot three feature vectors separately for each RoI $\mathbf{g},\mathbf{b_u},\mathbf{b_v}$.
>
> Q2. What is trained and frozen in the different tasks? For example in Figure 4, was the Encoder/L_triplet trained on just digits 4 or everything?
>
> In general, RoI encoder is fixed at all times. Yet for cMNIST experiments, we don't use any pretrained network, so we trained it in both ordinal embedding pretraining (stage 1) and agent training (stage 2). Projection head is trained in both stage 1 and 2. Agent is trained in stage 2 and 3 (adaptation). In stage 1 and 2, models are trained on the same source class, then being adapted to new target classes in stage 3.  Figure 4 follows the same strategy. More details in appendix A.1.
>
> Q3. Network and loss details are hidden in the text in random places, could you add a clear table in the Appendix for all of these?
>
> Please refer to Table 7 and Table 8 in appendix A.1 for more details.
>
> Q4. There is no mention of how the margin $m$ was chosen.
>
> We set margin as 60 heuristically.  It is not sensitive except in the selective localization experiment (Figure 3), where there are two different digits in each image. Please refer to appendix A.2 selective localization for training details and B.4 for a dedicated discussion regarding margin.
>
> Q5. ...why is it beneficial to treat this as a RL problem, instead of “just” outputting the whole sequence of actions in one go?
>
> By using RL,  the agent can leverage context information under a top-down paradigm.  The search strategy adaptively focuses on image regions containing objects by progressively discarding unrelated regions. Instead of outputting the whole sequences. The agent gathers information, stored in RNN, then decides the next step through an interactive way. Currently, the reward signal reflects small improvements in steps. We believe providing reward feedback to a sequence of actions directly shall be more challenging.
>
> Q6. ...treat the problem as a bandit setup. Could you discuss and contrast why that isn’t appropriate?
>
> It sounds like an interesting idea, but we are not quite familiar with the bandit setup. Could you clarify a little bit what is the idea of using this setup? Currently, given one state, the policy network learns to pick up one from a fixed set of actions.
>
> Q7. ...perturbed bounding boxes ...wasn’t extremely easy to find...It might be useful to spell this out clearly...did you find it to be sensitive?
>
> We reorganized Section 3.2 and now they are all in the same place. For the embedding distance-based reward to be effective and transferrable, the ordinal training needs to achieve high ordinal accuracy on both source and target domains. We do find the group box sampling scheme is more effective than the random sampling scheme.

---

> > ### Author Response · Authors · 2021-11-22
> > **Response -- Part 2**
> >
> > Q8. Figure 4 uses 50 images...Was the encoder trained on 50 images as well? Did you try using fewer?
> >
> > Yes, both RoI encoder, projection head and agent training on source class (digit 4) use 50 images. We've not tried fewer, as using 50 images to train an autoencoder is already hard. The model cannot reconstruct images very well when without data augmentation. So the bottleneck lies in the autoencoder, which the projection head is based on. Without the autoencoder, the ordinal embedding won't work well on this data.
> >
> > Q9. How dependent is this on the size of the examplar set?
> >
> > We have one such result reported in appendix B.6. Generally, it doesn't affect much in adaptation, probably because the data points in embedding space are compact.
> >
> > Q10. Did you explore other ways to use the exemplar set, instead of just averaging them into a prototype?
> >
> > This is an interesting idea! In the paper, we are following previous approaches, like prototypical network [1], to set the prototype as the mean vector of the embedded support points. Thanks for the suggestion! Prototype is not the only choice and we added some discussions in the paper.
> >
> > Q11. The “signed reward or not” section is not the clearest. ... “removing the sign operation” ...gives continuous rewards...
> >
> > We agree. Signed reward is popular in the RL literature, but in the context of evaluating object localization performances, it seems continuous reward provides more precise feedback and is easier to optimize than discrete ones.  However, both Tree-RL and active localization [2, 3]  use signed reward, probably because this is a tradition in RL research area where the environment is video game and the reward is win (+1) or loose (-1). We've changed 'with sign operation' to 'without sign operation' in the text. Sorry for the confusion.
> >
> > Q12. What is the loss_embed presented in Appendix A.2.1 and is it used for any results apart from the selective localisation one?
> >
> > $loss_{embed}$  simply means the total loss for training the ordinal embedding. In the two-digits scenario, in addition to reconstruction loss and triplet loss, we add another contrastive loss to encourage the separation of the center of the two digits. Thus the two centers are far away, and around each center, there is a local ordinal structure. When the center of one digit is within the ordinal structure of another digit, the agent will get confused and won't be able to find the target. This is why we set a very large margin in the contrastive loss.
> >
> > Reference:
> >
> > 1. Snell et al., Prototypical networks for few-shot learning, *NIPS 2017.*
> > 2. Juan C Caicedo et al., Active object localization with deep reinforcement learning, ICCV 2105.
> > 3. Zequn Jie et al., Tree-structured reinforcement learning for sequential object localization, NIPS 2016.

---

> > > ### Comment · Reviewer_yBs4 · 2021-12-01
> > > **Thank you for the response**
> > >
> > > I'd like to thank the authors for addressing most of my comments in much details, and improving their paper during this period.
> > >
> > > 1.  The modified Figures 1 and 2, tweaks to the main text and the greatly improved Appendix A help address my clarity concerns.
> > > 2. The work performed on Appendix B is very interesting and quite thorough. I think they provide more insights on the effects of individual components and provide very useful baselines. You might even want to bring some of these more to the main text or flag them more widely from the main paper, they are clearly interesting.
> > > 3. Sorry for being unclear about my point on the bandit setting. It really meant the point above in Q5, "make the agent output the whole sequence of actions in one go": Consider the following contextual bandit approach: given x, train a policy to directly output the whole sequence of actions/transformation as a variable-length list: [a_1, a_2, ..., a_T] to produce a final bounding box. The agent is only receiving the final reward for that final bounding box. The would remove any intermediate rewards for intermediate bounding boxes, hence a strictly more difficult setting (if the reward is helpful for these intermediate steps, which as you commented you believe it is) but with less sequential co-dependences and potential optimization complexities.
> > >
> > > Given the improvements to the paper I've increased my score to reflect this.

---

### Author Response · Authors · 2021-11-22
**Common Response**

We thank all reviewers for their constructive and encouraging feedback! We're glad that all the reviewers find our work novel and interesting. We highlight our main contributions and address the common concerns below.

Our work is positioned as an object localization method, which derives unknown bounding box locations from image class labels implicitly assumed known. Rather than proposing several candidate boxes, our approach outputs a single bounding box for the target object as the output, based on feature similarity to a limited set of query images. This is different from traditional detection work [1]. We propose a paradigm shift going from a class-agnostic, objectives-based approach to a class-specific,  feature similarity-based approach, and focus on the more challenging setting of adaptation with training on even one class. The problem we are solving is whether the location of the queried object can be found, given a class of images that belong to the same class.

**Evaluation protocol**

Both Reviewer tkLp and Reviewer DiRh asked about using CorLoc as the metric. CorLoc is widely used such as in the literature [2, 3] for evaluating object localization performance. Both reviewers mentioned the mAP metric - it is a more suitable metric for object detection, and similarly, recall is more suitable for evaluating object proposal generating approaches.

**Extension to detect multiple objects**

There are several dimensions in adding complexities to the query object localization problem. In this paper, we mainly focused on adapting a single object localizer to new tasks. The multiple/zero objects extension is another dimension to pursue. We agree with reviewer DiRh and reviewer tkLp that extensions to multiple/zero objects setting may not be trivial. It shall be handled very carefully and we have tuned down this claim in the revised paper. But we believe it shall be doable. For example, [1] proposed an approach in which the trained localizer is applied to the same image multiple times, and each time the objects already found are marked. A termination action can be added and it will not be triggered if the image contains no targeted objects. A multiple object extension of our approach can be proposed similarly.

**Off-the-shelf pre-trained networks**

Another contribution of our approach is the ordinal embedding-based reward signal. Reviewers asked questions about the advantages of ordinal embedding, compared with other off-the-shelf pre-trained networks. The ordinal reward signal has the advantage of being directly relevant to the downstream localization task, while also being highly transferable.  We added more experimental results to support these claims, including an ablation study on the proposed ordinal embedding (appendix B.1), comparison to a supervised baseline  (appendix B.2), and various choices of backbone networks (appendix B.3).

**Transfer learning vs. Few-shot setting**

We compared two orthogonal settings of training on the source domain: one-way many shots, and many-way few-shots. The question to answer is, in order to achieve good localization performance on a target domain, whether one needs to collect a few data from many source classes (as is often required in the few-shot setting), or extensively collect more data from a single source class (the transfer learning setting).  Experiments in 4.3 show that the proposed method can generalize well with one-way training data. It can achieve both objectives of specificity on the source tasks, and adaptability to target tasks.

**Clarity issue and summary of revision**

Reviewers provide many good suggestions on improving the clarity, such as adding the various missing details about the design choices and  experimental setup. We apologize for the confusion and in order to address these concerns, we have rewritten the methodology part in Section 3, reorganized the paper, and provided a full algorithmic pipeline (A.1) and implementation details (A.2) in the appendix.

The major revisions of the manuscript also include:

- Redraw of Figure 1
- Redraw of Figure 2
- Add Table 7 and Table 8 to summarize network and loss updating details
- Add a 2x2 ablation study on the ordinal and RL components  (appendix B.1)
- Add comparison to supervised method Faster RCNN (appendix B.2)
- Add comparison of several off-the-shelf backbone networks such as faster RCNN, ViT (appendix B.3)
- Add a discussion on the effects of margin parameter (appendix B.4)
- Add further comparison to few-shot training on data efficiency (appendix B.5)

1. Shaoqing Ren et al., Faster r-cnn: towards real-time object detection with region proposal networks, TPAMI 2016.
2. Deselaers et al., Weakly supervised localization and learning with generic knowledge, ICCV 2012.
3. Wei, Xiu-Shen, et al..  Unsupervised object discovery and co-localization by deep descriptor transformation, PR 2019.

---

### Decision · Program_Chairs · 2022-01-20

**Decision:**

Accept (Poster)

**Comment:**

The paper presents a reinforcement learning based approach for object localization given an exemplary set of images.  The paper shows that test-time policy adaptation to new environments is possible with object detection experiments. All four reviewers find the proposed approach interesting. Most of the reviewers feel their initial concerns (including the clarity of the paper and the approach details) addressed after the discussion with the authors and the revision. One reviewer still finds the experiments limited even after the author rebuttal, but the reviewers all agree that there is value in the paper.

We recommend accepting the paper.